# Structure of the HOPS tethering complex, a lysosomal membrane fusion machinery

Dmitry Shvarev[1†], Jannis Schoppe[2†], Caroline König[2†], Angela Perz[2], Nadia Füllbrunn[2], Stephan Kiontke[3], Lars Langemeyer[2], Dovile Januliene[1], Kilian Schnelle[1], Daniel Kümmel[4], Florian Fröhlich[5], Arne Moeller[1,6*], Christian Ungermann[2,6*]

[1]Department of Biology/Chemistry, Structural Biology section, Osnabrück University, Osnabrück, Germany; [2]Department of Biology/Chemistry, Biochemistry section, Osnabrück University, Osnabrück, Germany; [3]Department of Plant Physiology and Photo Biology, University of Marburg, Marburg, Germany; [4]Department of Chemistry and Pharmacy, Institute of Biochemistry, University of Münster, Münster, Germany; [5]Department of Biology/Chemistry, Molecular Membrane Biology group, Osnabrück University, Osnabrück, Germany; [6]Center of Cellular Nanoanalytic Osnabrück (CellNanOs), Osnabrück University, Osnabrück, Germany

**Abstract** Lysosomes are essential for cellular recycling, nutrient signaling, autophagy, and pathogenic bacteria and viruses invasion. Lysosomal fusion is fundamental to cell survival and requires HOPS, a conserved heterohexameric tethering complex. On the membranes to be fused, HOPS binds small membrane-associated GTPases and assembles SNAREs for fusion, but how the complex fulfills its function remained speculative. Here, we used cryo-electron microscopy to reveal the structure of HOPS. Unlike previously reported, significant flexibility of HOPS is confined to its extremities, where GTPase binding occurs. The SNARE-binding module is firmly attached to the core, therefore, ideally positioned between the membranes to catalyze fusion. Our data suggest a model for how HOPS fulfills its dual functionality of tethering and fusion and indicate why it is an essential part of the membrane fusion machinery.

**\*For correspondence:**
arne.moeller@uni-osnabrueck.de (AM);
cu@uos.de (CU)

†These authors contributed equally to this work

**Competing interest:** The authors declare that no competing interests exist.

## Editor's evaluation

This landmark study reports the cryo-EM structure of HOPS, a heterohexameric tether that participates in the fusion of late endosomes, autophagosomes, and AP-3 vesicles with lysosomes. The structure provides a convincing update of earlier, lower-resolution models. Interestingly, the SNARE-binding module is attached to the core of the complex. These results suggest possible mechanisms by which HOPS could catalyze SNARE-dependent fusion.

## Introduction

Lysosomal fusion underlies a plethora of cellular processes. It is essential in the maintenance and upkeep of eukaryotic membranes and fundamental to secretion, endocytosis, and autophagy (*Saftig and Puertollano, 2021*). Macromolecules from different trafficking pathways end up in lysosomes where they are degraded (*Klionsky et al., 2021*; *Saftig and Puertollano, 2021*). This process relies on multiple fusion events within the endomembrane system. In general, fusion depends on SNAREs, which are present on opposite membranes and zipper into four-helix bundles with the help of Sec1/Munc1 (SM) proteins (*Jahn and Fasshauer, 2012*; *Südhof and Rothman, 2009*; *Wickner and Rizo, 2017*). Each assembled SNARE complex contains three Q-SNAREs ($Q_a$, $Q_b$, and $Q_c$) and one R-SNARE,

**eLife digest** Our cells break down the nutrients that they receive from the body to create the building blocks needed to keep us alive. This is done by compartments called lysosomes that are filled with a cocktail of proteins called enzymes, which speed up the breakdown process. Lysosomes are surrounded by a membrane, a barrier of fatty molecules that protects the rest of the cell from being digested. When new nutrients reach the cell, they travel to the lysosome packaged in vesicles, which have their own fatty membrane. To allow the nutrients to enter the lysosome without creating a leak, the membranes of the vesicles and the lysosome must fuse.

The mechanism through which these membranes fuse is not fully clear. It is known that both fusing membranes must contain proteins called SNAREs, which wind around each other when they interact. However, this alone is not enough. Other proteins are also required to tether the membranes together before they fuse. To understand how these tethers play a role, Shvarev, Schoppe, König et al. studied the structure of the HOPS complex from yeast. This assembly of six proteins is vital for lysosomal fusion and, has a composition similar to the equivalent complex in humans.

Using cryo-electron microscopy, a technique that relies on freezing purified proteins to image them with an electron microscope and reveal their structure, allowed Shvarev, Schoppe, König et al. to provide a model for how HOPS interacts with SNAREs and membranes. In addition to HOPS acting as a tether to bring the membranes together, it can also bind directly to SNAREs. This creates a bridge that allows the proteins to wrap around each other, driving the membranes to fuse.

HOPS is a crucial component in the cellular machinery, and mutations in the complex can cause devastating neurological defects. The complex is also targeted by viruses – such as SARS-CoV-2 – that manipulate HOPS to reduce its activity. Shvarev, Schoppe, König et al.'s findings could help researchers to develop drugs to maintain or recover the activity of HOPS. However, this will require additional information about its structure and how the complex acts in the biological environment of the cell.

---

which are categorized according to the interaction of the glutamine and arginine residues in the central hydrophilic layer of the otherwise hydrophobic interfaces within the SNARE complex (*Jahn and Fasshauer, 2012*; *Südhof and Rothman, 2009*; *Wickner and Rizo, 2017*). Prior to fusion, specialized tethering complexes establish tight links between organelles and interact with SM proteins to promote fusion (*Baker and Hughson, 2016*; *Kuhlee et al., 2015*; *Ungermann and Kümmel, 2019*). Despite their central position in trafficking, the underlying mechanics of tethering complexes and how they catalyze membrane fusion remain unresolved.

The heterohexameric HOPS complex mediates fusion of late endosomes, autophagosomes, and AP-3 vesicles with mammalian lysosomes or vacuoles in yeast (*van der Beek et al., 2019*; *Spang, 2016*; *Wickner and Rizo, 2017*), and is probably the best-studied tethering complex. Fusion assays using yeast vacuoles or reconstituted SNARE-bearing proteoliposomes showed that HOPS is essential for membrane fusion at physiological SNARE concentrations (*D'Agostino et al., 2017*; *Mima et al., 2008*; *Zick and Wickner, 2016*). HOPS is the target of viruses such as SARS-CoV2 (*Miao et al., 2021*), and its inactivation blocks Ebola infections (*Carette et al., 2011*). Furthermore, multiple HOPS mutations can cause severe diseases ranging from Parkinson's to lysosomal disorders (*van der Beek et al., 2019*; *Sanderson et al., 2021*; *van der Welle et al., 2021*).

Five out of six HOPS subunits (Vps11, Vps16, Vps18, Vps39, and Vps41) are predicted to share a similar architecture, comprising an N-terminal β-propeller and a C-terminal α-solenoid domain (*Figure 1A*). Vps11 and Vps18 form the core and carry conserved C-terminal RING finger domains (*Rieder and Emr, 1997*), which are essential for HOPS formation (*Hunter et al., 2017*), but also show E3 ligase activity on their own (*Segala et al., 2019*). At the opposite sites, Vps41 and Vps39 bind to membrane-anchored small GTPases (the Rab7-like Ypt7 in yeast) (*Bröcker et al., 2012*; *Lürick et al., 2017*; *Ostrowicz et al., 2010*), while Vps16 and the SM protein Vps33 establish the SNARE-binding module (*Baker et al., 2015*; *Graham et al., 2013*). Low-resolution negative-stain electron microscopy (EM) analyses revealed the overall arrangement of HOPS (*Bröcker et al., 2012*; *Chou et al., 2016*), yet were insufficient to localize the exact position of its subunits and suggested significant flexibility within the particle.

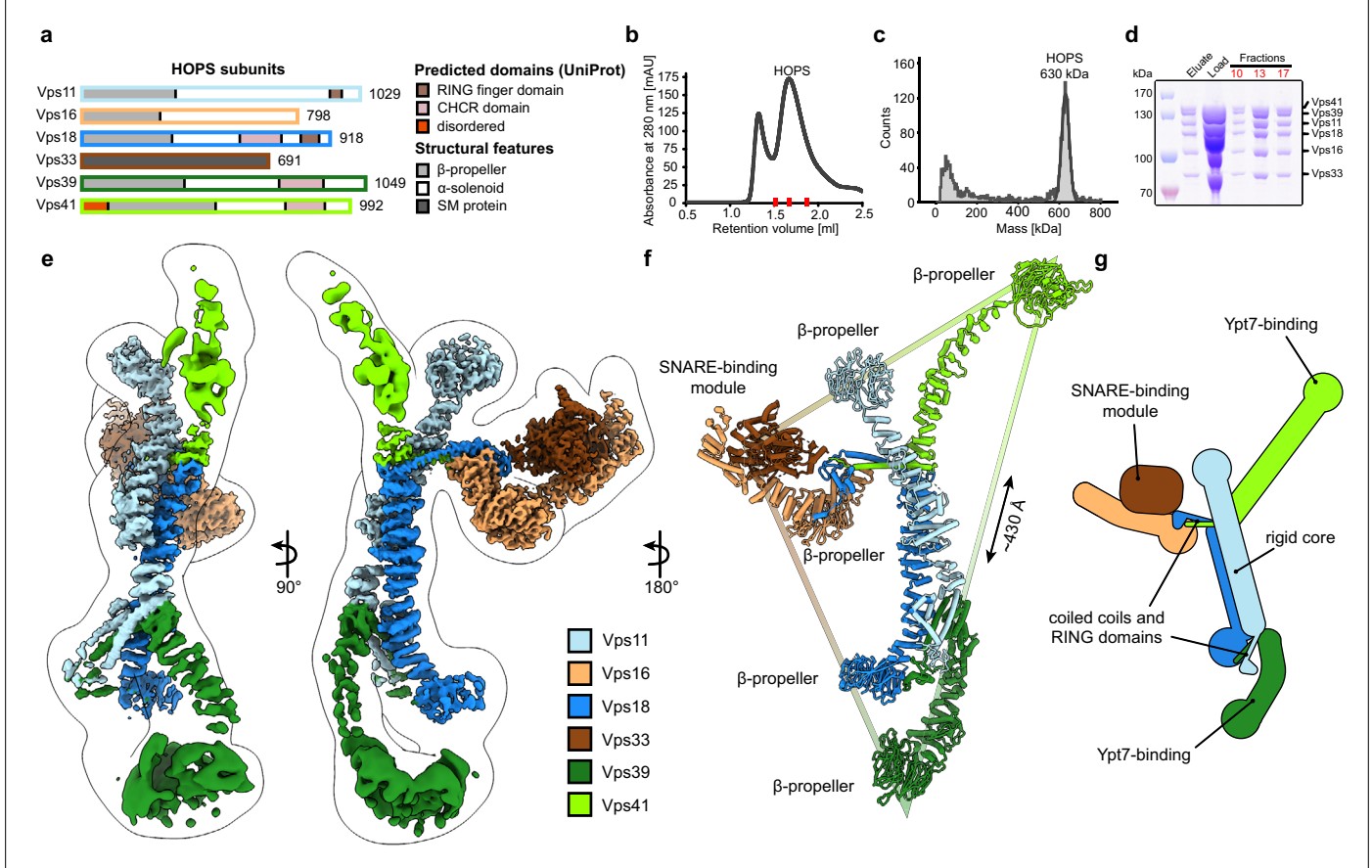

**Figure 1.** Composition and architecture of the yeast HOPS complex. (**A**) Domain architecture and size of HOPS subunits. Predicted domains and structural features are indicated. (**B**) Size exclusion chromatography (SEC) of the affinity-purified HOPS. Purification was done as described in Materials and methods. (**C**) Mass photometry analysis. Peak fractions from SEC were analyzed for size. (**D**) Purified HOPS. Proteins from affinity purification (eluate) and SEC (red dashes in (**B**)) were analyzed by SDS-PAGE. (**E**) Overall architecture of the HOPS complex. Composite map from local refinement maps (Figure S1-3) was colored by assigned subunits. One of the consensus maps used for local refinements was low-pass-filtered and is shown as a transparent envelope. (**F**) Atomic model of the HOPS complex. For the N-terminal fragments of Vps41 and Vps39, which were not resolved to high resolution by local refinements, AlphaFold models are used and manually fitted into the densities of consensus maps (*Figure 1—figure supplements 1 and 2*). The triangular shape of the complex is highlighted with the approximate distance between the β-propellers of Vps41 and Vps39. (**G**) Schematic representation of the HOPS complex indicates central features.

The online version of this article includes the following source data and figure supplement(s) for figure 1:

**Source data 1.** Gels and graphs for *Figure 1b, c and d*.

**Figure supplement 1.** Cryo-EM data processing workflow.

**Figure supplement 2.** Consensus maps and corresponding local refinement maps.

**Figure supplement 3.** Cryo-EM analysis of HOPS.

**Figure supplement 4.** Biochemical analysis of HOPS mutants lacking N-terminal β-propellers.

**Figure supplement 4—source data 1.** Blots and gels corresponding the experiments.

The way HOPS fulfills its function remains speculative, and multiple mechanisms have been proposed, including a role as a bulky membrane stressor (*D'Agostino et al., 2017*) or, conversely, as a highly flexible membrane tether (*Bröcker et al., 2012*; *Chou et al., 2016*). In the absence of detailed structural data, it remains obscure how HOPS facilitates lysosomal fusion.

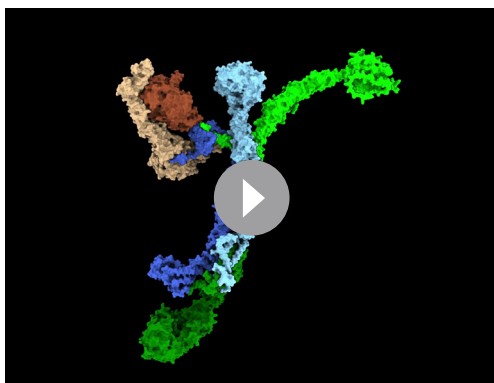

**Video 1.** Overall architecture of HOPS tethering complex: transition between ribbon and molecular surface representation.
https://elifesciences.org/articles/80901/figures#video1

## Results

### Structure of the HOPS complex

Previously, structural studies were hampered by the low stability and flexibility of the complex, which required fixation through mild crosslinking for sample preparation and confined structural studies to negative stain analyses (*Bröcker et al., 2012*; *Chou et al., 2016*). To enable high-resolution cryo-EM of non-crosslinked HOPS, we vastly improved and accelerated our purification protocols and removed any delays during the sample preparation procedure (*Figure 1B–D*). Single-particle analysis including extensive classifications followed by local refinements led to a composite structure with resolutions between 3.6 and 5 Å (*Figure 1E* and *Figure 1—figure supplements 1–3*).

HOPS forms a largely extended, slender structure extending approximately 430 Å in height and 130 Å in width, resembling a triangular shape (*Figure 1E–G*, *Video 1*). In the center of the modular complex, Vps11 and Vps18 align antiparallel through their elongated α-solenoids, establishing a large interface area of 1972 Å$^2$ (*Figure 1E-G*, *Figure 2*, *Figure 2—figure supplement 4*), which resulted in the highest resolution obtained within HOPS (*Figure 1—figure supplement 3*) and is comparable with protein interfaces in other complexes with similar structural elements as in HOPS (e.g. *Kschonsak et al., 2022*). Interestingly, AlphaFold predicts a long unstructured region within Vps11 (Q760 to D784), resulting in an upper and lower part of the subunit. However, this region is clearly resolved and organized in our density. In our model, the two core subunits create a central assembly hub for the four other subunits (Vps39, Vps41, Vps16, and Vps33) that fulfill specific functions and localize to the periphery of the complex. The N-termini of Vps11 and Vps18 are located distally from the core and each carries a β-propeller, which can be deleted without affecting the complex formation (*Figure 3A and D*, *Figure 1—figure supplement 4*). At the C-termini, both Vps11 and Vps18 have long α-helices which are followed by RING finger domains (*Figures 1E–G, 2A and B*, *Figure 2—figure supplement 1*). Both features are key elements for the stability of the modular architecture and serve as anchor points for the additional subunits. In agreement, HOPS, carrying mutations in the RING finger domains of Vps11 (*vps11-1*) (*Peterson and Emr, 2001*), selectively lost Vps39, whereas only Vps41 was obtained from a similar Vps18 mutant.

Vps41 and Vps39 provide the Ypt7-interaction sites at their peripheral N-terminal regions. Their extended C-terminal helices, similar to those of Vps11 and Vps18, are tightly interlocked through coiled-coil motifs with the long C-terminal α-helices and RING finger domains from the respective core subunits (Vps41 with Vps18, and Vps39 with Vps11) (*Figure 2A and B*). Additional stability of Vps39 within the complex is provided by the interaction of the long α-helix at the C-terminus of Vps39 with the β-propeller of Vps18 (*Figure 2D*). In our density, peripheral portions of both Vps41 and Vps39 are least well resolved indicating their considerable flexibility. Multi-modular classification analyses revealed angular re-orientations of about 9° for Vps41 and 20° for Vps39 (*Figure 2—figure supplement 3A-D*) relative to the core, resulting in variable positions of the N-terminal β-propellers. At the top, Vps41 reaches out by approximately 100 Å in length and, similarly, Vps39 forms an elongated arch at the bottom (*Figure 1E–G*), positioning both Ypt7-interacting units at the farthest ends of the complex.

The SNARE-binding element, composed of Vps16 and the SM protein Vps33, branches out to the lateral side of the complex approximately at the center of the structure (*Figures 1E–G and 2*). Vps16 shares a large interface with the coiled-coil motif formed by Vps18 and Vps41 and the N-terminus of Vps18, which is stabilized through interactions between hydrophobic and charged residues (*Figure 2C*). Vps33 is in immediate contact with the structured loop of Vps18 (residues 824–831) that connects the elongated helix with the RING finger domain (*Figure 2C*). This, as well as the role of RING finger domains in the interlocking of other subunits, explains why mutations at RING domains result in devastating human diseases and HOPS dysfunction (*van der Beek et al., 2019*; *Edvardson*

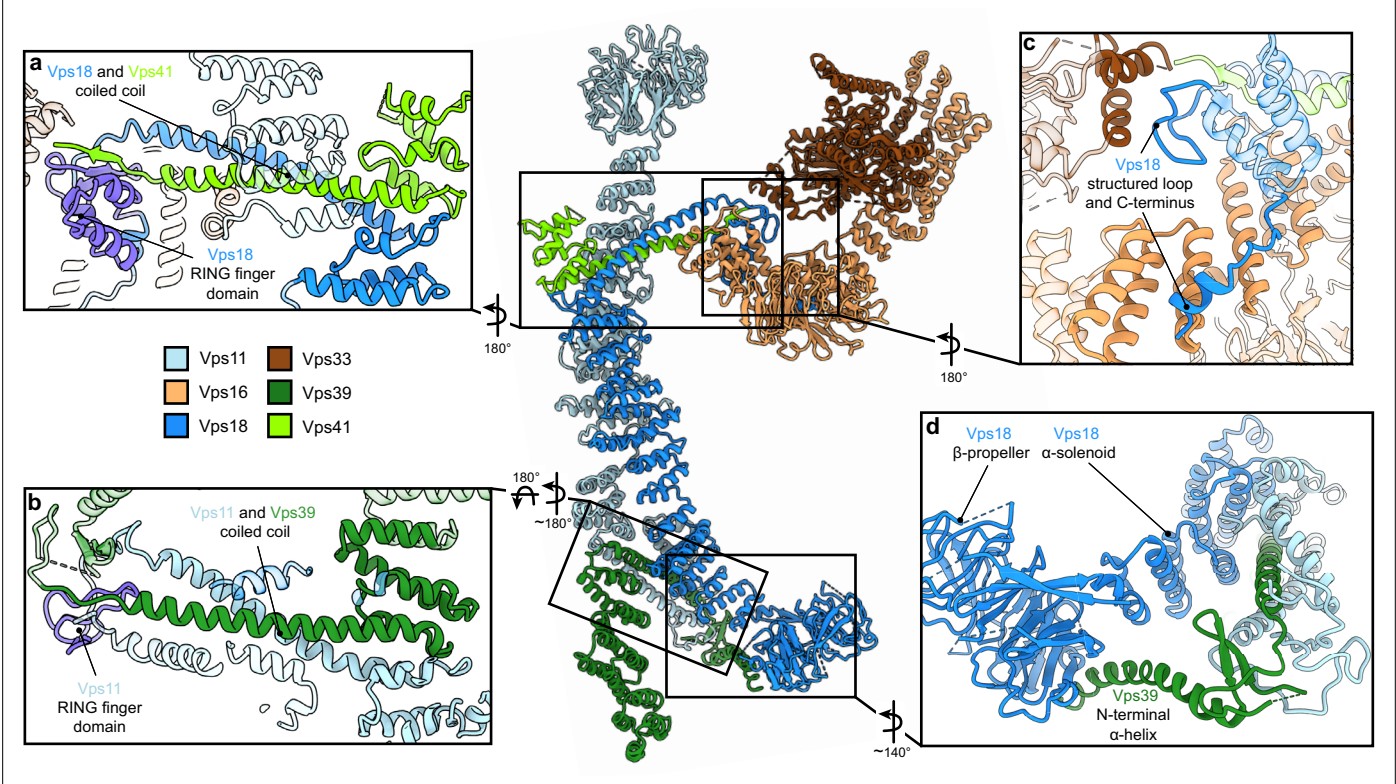

**Figure 2.** Vps11 and Vps18 C-termini as central interaction hubs for all other subunits. Atomic model of HOPS with highlighted interaction sites between subunits. (**A, B**) Coiled-coil motifs followed by the RING finger domains (violet) are the key structural features of HOPS. (**A**) The Vps18 C-terminal hub. Vps18 and Vps41 interact via the coiled coil and the Vps18 RING finger domain (displayed as non-transparent cartoons). (**B**) The Vps11 C-terminal hub. Vps11 interacts via its RING finger domain and the coiled coil with Vps39 (displayed as non-transparent cartoons). (**C**) Connection of the SNARE binding module (Vps33 and Vps16) to the backbone of HOPS via interactions with the structured loop at the RING finger domain and the C-terminus of Vps18 (displayed as non-transparent cartoons). (**D**) Vps39 connects by its C-terminal helix the β-propeller of Vps18, which provides additional stability in this part of HOPS.

The online version of this article includes the following source data and figure supplement(s) for figure 2:

**Figure supplement 1.** Comparison of Vps11 and Vps18 RING finger domains.

**Figure supplement 2.** Key role of the RING finger domains of Vps11 and Vps18 in HOPS stability.

**Figure supplement 2—source data 1.** Mass spectrometry raw data.

**Figure supplement 3.** Flexibility analysis of the HOPS complex.

**Figure supplement 4.** Interactions between core subunits Vps11 and Vps18.

*et al., 2015*; *Robinson et al., 1991*; *van der Welle et al., 2021*; *Zhang et al., 2016*) and cause failure of correct HOPS assembly (*Figure 2—figure supplement 2*). Overall, the SNARE binding module appears to be stably connected to the central core, while only the short C-terminal section of Vps16 α-solenoid (residues 739–798) displays high variability and is not resolved in our structure.

## HOPS couples tethering to fusion activity

Tethering complexes bridge membranes by binding small GTPases, but also harbor or bind SM proteins (*Ungermann and Kümmel, 2019*). Reconstituted vacuole fusion is strictly HOPS and Ypt7-dependent at physiological SNARE concentrations (*Langemeyer et al., 2018*; *Zick and Wickner, 2016*), suggesting that HOPS is not just a tether, but part of the fusion machinery (*Baker et al., 2015*; *Wickner and Rizo, 2017*). However, so far, it was unknown how tethering and fusion activities of HOPS may be linked mechanistically. To address this, we first analyzed the N-terminal β-propellers of Vps41 and Vps39, the likely binding sites with Ypt7 (*Lürick et al., 2017*; *Ostrowicz et al., 2010*; *Plemel et al., 2011*). The intrinsic low affinity between HOPS and Ypt7 (*Lürick et al., 2017*) prevented reconstitution of the complex for structural studies, therefore, we instead relied on AlphaFold predictions.

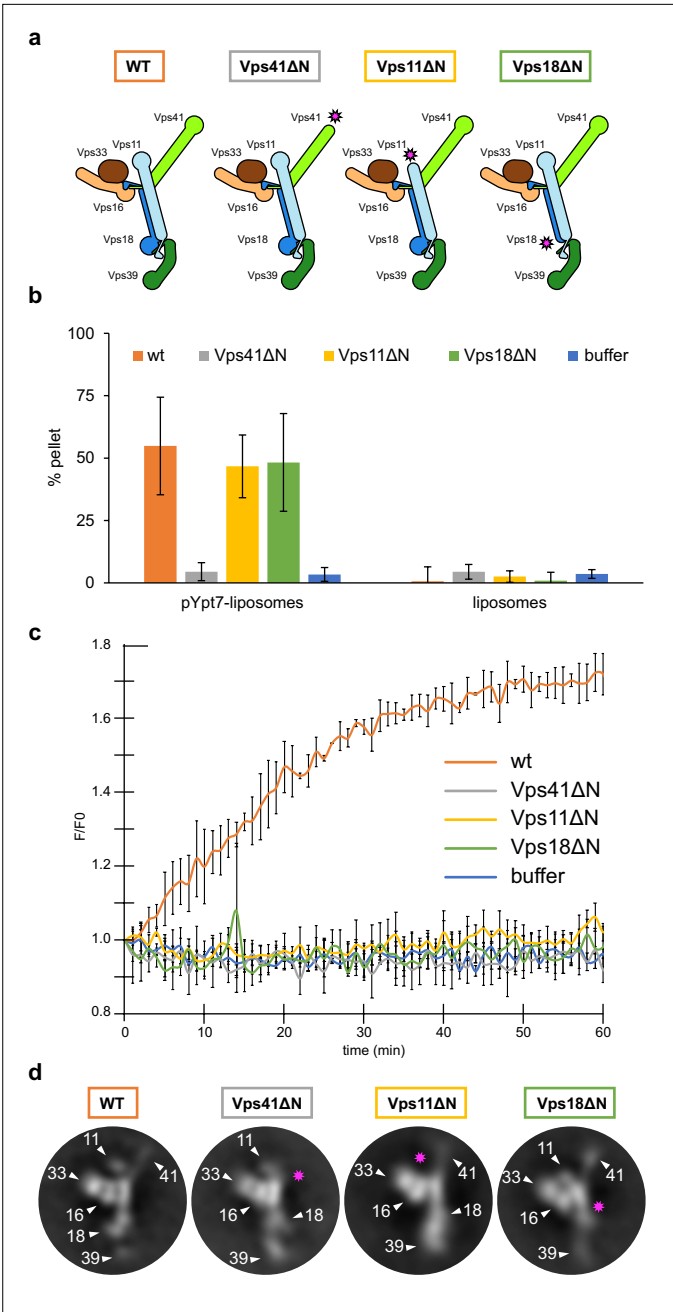

**Figure 3.** HOPS couples tethering and fusion activities. (**A**) Schematic representations of HOPS wild-type (as in *Figure 1G*) and mutants lacking N-terminal β-propeller domains (indicated by pink asterisks). (**B**) Tethering assay. Fluorescently labeled liposomes loaded with prenylated Ypt7-GTP or none were incubated with HOPS and mutant complexes. Tethering was determined as described in Materials and methods. Data shown from three biological replicates, bars indicate standard deviation. (**C**) Fusion assay. Fusion of proteoliposomes carrying vacuolar SNAREs were preincubated with Ypt7-GDI, GTP, and Mon1-Ccz1. For fusion, HOPS wild-type or mutant and the soluble Vam7 SNARE were added (*Langemeyer et al., 2020*; *Langemeyer et al., 2018*). Analysis was done as described (*Zick and Wickner, 2016*). See Materials and methods. Data shown from three biological replicates, bars indicate standard deviation. (**D**) Representative 2D class averages obtained from negative-stain analyses of wild-type HOPS and mutants. Pink asterisks indicate missing densities in the mutants.

The online version of this article includes the following source data and figure supplement(s) for figure 3:

**Figure supplement 1.** Ypt7 interaction with Vps41 and Vps39.

**Source data 1.** Raw data for tethering (*Figure 3B*) and fusion assay (*Figure 3C*).

Additionally, we solved the structure of the β-propeller of *Chaetomium thermophilum* Vps39 by X-ray crystallography, which largely confirmed the predicted model (*Figure 3—figure supplement 1A*). Surprisingly, in the AlphaFold model, Ypt7 binding occurs at the α-solenoid of Vps39 where it does not directly interact with the β-propeller (*Figure 3—figure supplement 1C*), as originally expected. Furthermore, the binding site on Vps39 is placed approximately 5–6 nm above the membrane if Ypt7-anchored HOPS is in an upright position on supported lipid bilayers (*Füllbrunn et al., 2021*). Membrane-bound Ypt7 can still reach this site due to its 10 nm long hypervariable domain (not shown in the prediction).

In the predicted complex of Vps41 (residues 1–919) with Ypt7 (residues 1–185), the GTPase binds directly to the Vps41 β-propeller, as anticipated. However, it interacts on the opposite side from the membrane-interacting amphipathic lipid-packing sensor (ALPS) motif (*Cabrera et al., 2010*; *Figure 3—figure supplement 1B*), suggesting that the hypervariable region of Ypt7 is required for binding, in analogy to Vps39. Curiously, in the predicted model, the ALPS motif faces away from the membrane which would hamper membrane binding. We noted, however, that this distal region of Vps41 (*Figure 3—figure supplement 1B*) displays substantial flexibility. The β-propeller of Vps41 might, therefore, be oriented differently in the structure than predicted by AlphaFold, which may bring the ALPS motif in contact with the bilayer if Vps41 is bound to Ypt7. Nevertheless, future experimental data will need to confirm the predicted AlphaFold model of Vps41 and Vps39 interaction with Ypt7.

Vps39 binds Ypt7 far stronger than Vps41 (*Auffarth et al., 2014*; *Lürick et al., 2017*), and may be assisted by Vps18 to sandwich Ypt7, whereas Vps41 binds to Ypt7 apparently alone (*Figure 3—figure supplement 1D*). Such a dual interaction could explain both tighter binding and a preferred orientation of HOPS on membranes (*Füllbrunn et al., 2021*). To test the functional importance, we generated HOPS complexes lacking the β-propeller of Vps11, Vps18, or Vps41 (*Figure 3A*). All complexes were purified in equimolar stoichiometry (*Figure 1—figure supplement 4A*), and interacted with Ypt7, but not the Golgi Rab Ypt1 in GST pull-down assay, suggesting that at least one Rab-binding site is maintained in all truncated complexes (*Bröcker et al., 2012*; *Lürick et al., 2017*; *Ostrowicz et al., 2010*; *Zick and Wickner, 2016*; *Zick and Wickner, 2012*; *Figure 1—figure supplement 4B*).

To determine the activity of HOPS mutants, we compared tethering and fusion. For tethering, we incubated liposomes bearing Ypt7-GTP with each complex and quantified clustering after centrifugation (*Füllbrunn et al., 2021*; *Lürick et al., 2017*). HOPS lacking the Vps41 β-propeller was inactive as shown (*Lürick et al., 2017*), whereas HOPS with truncated Vps11 or Vps18 was fully functional and as efficient as wild-type HOPS (*Figure 3B*). In contrast, when added to SNARE and Ypt7-GTP bearing

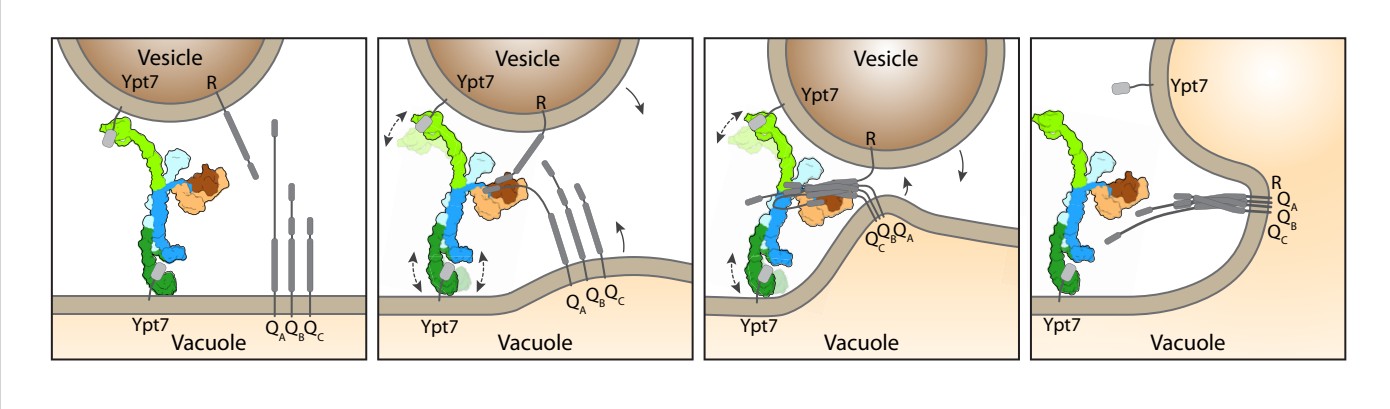

**Figure 4.** Working model for HOPS-mediated membrane tethering and fusion. The HOPS complex binds to Ypt7 on the vacuole and vesicles via Vps39 (dark green) and Vps41 (light green). SNARE proteins are recruited to HOPS by their N-terminal domains and the SNARE-binding module (dark and light brown). The stable central core of HOPS keeps the membranes in place, while Vps41 and Vps39 may function as dampers due to their limited flexibility. Consequentially, zippering of SNAREs, which is initiated by binding to Vps33 (dark brown), begins. As the N-terminal domains of SNAREs bind to HOPS, further SNARE zippering may occur with the HOPS backbone acting as a lever (not shown here). This may cause membrane stress and thus catalyzes fusion. HOPS may let go of Ypt7 and SNAREs thereafter. For details see text.

The online version of this article includes the following figure supplement(s) for figure 4:

**Figure supplement 1.** Point mutations affecting HOPS function.

liposomes, only wild-type HOPS, but none of the mutant complexes promoted fusion (*Figure 3C*). This was particularly puzzling for HOPS lacking either the Vps18 or Vps11 β-propeller as they had full tethering activity (*Figure 3B*). Therefore, we compared the structural features of HOPS mutants with the wild type using negative stain EM. Deletions of the β-propellers of Vps41, Vps11, or Vps18 indeed resulted in a loss of protein density at the expected positions, while preserving the densities of all other subunits (*Figure 3D*). Interestingly, HOPS complexes lacking β-propellers in Vps11 or Vps18 showed an alteration in the relative orientation of Vps39 within the complex in some 2D class averages. This observed structural variation might be a result of the increased flexibility of mutant HOPS due to the lack of structural support by β-propellers of Vps18 and Vps11. We conclude that the β-propellers of Vps18 and Vps11 contribute to the overall structure of the HOPS complex or may play a stabilizing role during the fusion process, which would explain, why they are essential for the full activity of HOPS.

## Discussion

Our data suggest a working model of how HOPS catalyzes fusion at lysosomes and vacuoles (*Figure 4*). According to our structure, the three major ligand binding sites of HOPS are arranged in a triangular fashion. While the two Ypt7-binding sites show significant conformational variability, the SNARE-interacting unit is firmly connected to the stable backbone formed by Vps11 and Vps18. For tethering, HOPS Vps39 and Vps41 bind Ypt7 on target membranes. During this process, HOPS remains upright on membranes (*Füllbrunn et al., 2021*). Then, the SM protein Vps33 and possibly other sites on HOPS (*Krämer and Ungermann, 2011*; *Baker et al., 2015*; *Lürick et al., 2017*; *Lürick et al., 2015*; *Song et al., 2020*) bind SNAREs from the opposing membranes, and zipper them up toward their membrane anchor (*Figure 4*). Note, that this process can be blocked by Orf3a in the COVID-19 SARS-CoV-2 virus (*Miao et al., 2021*). In our model, the backbone of HOPS dampens the movement of the vesicles and acts as a lever arm holding on to SNAREs during zippering (*D'Agostino et al., 2017*). This three-point arrangement would cause membrane stress and could explain how HOPS catalyzes membrane fusion (*Figure 4*). The physiological function of HOPS can be bypassed if large complexes are redirected to SNAREs at the fusion site (*D'Agostino et al., 2017*; *Song et al., 2017*), which can even promote fusion of deficient SNARE complexes (*Orr et al., 2022*; *Song et al., 2021*). Zippered SNAREs may then dissociate from HOPS and allow access for α-SNAP and NSF to recycle SNAREs (*Zhang and Hughson, 2021*).

How tethering complexes contribute to fusion poses a long-standing question in the field. The process necessitates the binding of two opposing membranes and exact coordination of the zippering procedure of membrane-bound SNAREs. Previous analyses suggested strong flexibility along the entire HOPS structure, which was interpreted as a hallmark of tethering complexes and an essential prerequisite to their function (*Bröcker et al., 2012*; *Chou et al., 2016*; *Füllbrunn et al., 2021*; *Ha et al., 2016*). Structural models of a largely open HOPS, based on negative stain EM. *Chou et al., 2016*, supported the importance of HOPS structural flexibility. Flexibility was also observed in our previous negative stain EM structure (*Bröcker et al., 2012*). Instead, our cryo-EM data, collected on a highly pure and homogeneous sample that was not modified by any crosslinker or other fixative agents like negative stain, show that HOPS flexibility is limited. Taking the obtained resolution as a measure of sample flexibility (*Rawson et al., 2016*), we conclude that the backbone and the SNARE-binding module are the least flexible parts of HOPS and appear to be stably associated with each other. In contrast, the membrane interacting units of Vps41 and Vps39 show some flexibility with 10–20° movements between particles, which substantially reduced their resolution (*Figure 2—figure supplement 3*). We relate this flexibility to their function within HOPS, where it may dampen the motion of HOPS between membranes and stabilizes the SNARE interaction, which are necessary for fusion (*Laage and Ungermann, 2001*; *Krämer and Ungermann, 2011*; *Lürick et al., 2015*; *Baker et al., 2015*; *Song et al., 2020*). Mammalian HOPS contains the same six subunits as yeast HOPS, which are generally highly conserved and will therefore function similarly (*van der Beek et al., 2019*; *van der Kant et al., 2015*). Our structure can hence be used to map disease prone mutations (*Figure 4—figure supplement 1*) and may thus explain the consequences on HOPS function.

Due to low binding affinities, no structures of tethering complexes bound to Rabs have been solved. Here, we used AlphaFold to predict the interfaces of HOPS with its bound Rab7-like Ypt7 protein to understand the positioning of HOPS during tethering (*Figure 3—figure supplement 1*).

We suspect that this low binding affinity helps tethering complexes to also let go of the Rab when fusion progresses (*Figure 4*). Even though long coiled-coil tethers such as EEA1 can promote SNARE-mediated membrane fusion (*Murray et al., 2016*), their cooperation with Rab GTPases in fusion is likely quite different from the mechanism proposed here (*Ungermann and Kümmel, 2019*).

Some tethering complexes such as CORVET, CHEVI, or FERARI have attached SM proteins (*van der Beek et al., 2019*; *Solinger et al., 2020*), others such as COG, GARP, Exocyst, and Dsl cooperate with SM proteins (*Ungermann and Kümmel, 2019*), which may catalyze fusion similar to HOPS. For each of these complexes, in vivo models for their function exist, yet proteoliposome fusion assays in the presence of the required small GTPases are either not available or not yet completely developed (*Balderhaar et al., 2013*; *Ha et al., 2016*; *Lee et al., 2022*; *Maib and Murray, 2022*; *Ren et al., 2009*; *Rossi et al., 2020*; *Solinger et al., 2020*). We expect that similar approaches as established for HOPS will further support the key role of tethering complexes and reveal their cooperation with SM proteins in SNARE-mediated membrane fusion.

The overarching principle suggested here for HOPS is not limited to lysosomal fusion but may extend to synaptic transmission, where the tether Munc13 and the SM protein Munc18 cooperatively catalyze the N- to C-terminal zippering of SNAREs during fusion (*Lai et al., 2017*; *Rizo, 2022*; *Stepien et al., 2022*; *Stepien and Rizo, 2021*). Both Munc18 and Vps33 interact similarly with the N-terminal part of the SNARE domains of the R- and Qa-SNARE and may promote assembly until the central zero-layer of the SNARE domain (*Baker et al., 2015*; *Stepien et al., 2022*).

During HOPS-mediated fusion, SNARE zippering beyond the zero-layer could then proceed, while the Vps33 lets go of the forming four-helix bundles (*Stepien et al., 2022*). However, HOPS binds both the N-terminal extensions of the Qa-SNARE Vam3 and other SNAREs, possibly via different binding sites along the HOPS complex (*Laage and Ungermann, 2001*; *Lürick et al., 2015*; *Song et al., 2020*). This association may thus maintain the force on membranes to catalyze full fusion, even if the SM protein lets go of the assembling SNARE complex (*Figure 4*). In agreement, HOPS complexes with deficient Vps33 can catalyze fusion of proteoliposomes only if the SNARE density is high (*Baker et al., 2015*). In turn, vacuoles expressing a Qa-SNARE Vam3 variant lacking the N-terminal extension, which is needed to bind HOPS, show diminished fusion (*Laage and Ungermann, 2001*). This suggests that HOPS supports SNAREs by templating the association of R- and Qa-SNAREs and by binding the N-terminal regions of SNAREs.

We believe that the deletion of the β-propeller in Vps11 or Vps18 result in a similarly deficient HOPS due to a less stable backbone (*Figure 3*). In either case, coupling between a stable backbone of HOPS and SNARE binding may be impaired and could result in less specific activity as fusion catalysts. Future experiments are required to determine the precise reason for their fusion deficiency.

Overall, our insights provide a novel blueprint to understand HOPS function, dynamics, and regulation both in fusion and in other functions of its subunits (*van der Beek et al., 2019*; *Elbaz-Alon et al., 2014*; *Hönscher et al., 2014*; *González Montoro et al., 2021*; *González Montoro et al., 2018*; *Wong et al., 2020*), and imply a general role of tethering complexes as a catalytic part of the fusion machinery.

## Materials and methods

### Yeast strains

Yeast strains used in this study are listed in *Supplementary file 1*. In general, HOPS subunits were expressed under the control of the GAL1 promoter according to the standard protocol (*Janke et al., 2004*). For HOPS subunit truncation (Vps41, Vps11, and Vps18) of the N-terminal part, the GAL1 promotor was inserted into the genome at the respective position. The 3x-FLAG Tag was attached to the HOPS subunit Vps41, except for the 18 ΔN mutant.

### Protein expression and purification from *Escherichia coli*

Rab GTPases for pulldown or tethering and fusion assays were expressed in *Escherichia coli* BL21 (DE3) Rosetta cells. Cells were grown in Luria broth (LB) medium complemented with 35 µg/ml kanamycin and 30 µg/ml chloramphenicol. Cultures were induced at $OD_{600}=0.6$ with 0.5 mM isopropyl-β-d-thiogalactoside (IPTG) overnight at 16°C before harvesting by centrifugation ($4800 \times g$, 10 min, 4°C). Cells resuspended in buffer (150 mM NaCl, 50 mM HEPES/NaOH, pH 7.4, 10 % glycerol, 1 mM PMSF,

and 0.5-fold protease inhibitor mixture [PIC]) were lysed by a Microfluidizer (Microfluidics Inc) and centrifugated at 25,000×*g*, 30 min, 4°C. Supernatants were incubated with glutathione Sepharose (GSH) fast flow beads (GE Healthcare) for GST-tagged proteins or nickel–nitriloacetic acid (Ni-NTA) agarose (Qiagen) for His-tagged proteins for 2 hr at 4°C. The proteins were eluted with buffer (150 mM NaCl, 50 mM HEPES/NaOH, pH 7.4, 10 % glycerol) containing 25 mM glutathione or 300 mM imidazole. Buffer was exchanged via a PD10 column (GE Healthcare). For tag cleavage, TEV protease was added after washing and incubated overnight. All proteins were stored at −80°C.

## Purification of the 3xFLAG-tagged HOPS complex variants from yeast

Two liters of yeast peptone (YP) medium containing 2 % galactose (v/v) were inoculated with 6 ml of an overnight culture. Cells were grown for 24 hr and harvested by centrifugation (4800×*g*, 10 min, 4°C). Pellets were washed with cold HOPS purification buffer (HPB, 1 M NaCl, 20 mM HEPES/NaOH, pH 7.4, 1.5 mM MgCl$_2$, and 5% (v/v) glycerol). The pellet was resuspended in a 1:1 ratio (w/v) in HPB supplemented with 1 mM phenylmethylsulfonylfluoride (PMSF), 1× FY protease inhibitor mix (Serva) and 1 mM dithiothreitol (AppliChem GmbH) and afterward dropwise frozen in liquid nitrogen before being lysed in a freezer mill cooled with liquid nitrogen (SPEX SamplePrep LLC). The powder was thawed on ice and resuspended in HPB supplemented with 1 mM PMSF, 1× FY, and 1 mM DTT using a glass pipette, followed by two centrifugation steps at 5000 and 15,000×*g* at 4°C for 10 and 20 min, respectively. After centrifugation, the supernatant was added to 2 ml of anti-FLAG M2 affinity gel (Sigma-Aldrich) and gently agitated for 45 min at 4°C on a nutator. Beads were briefly centrifuged (500×*g*, 1 min, 4°C) and the supernatant was removed. Beads were transferred to a 2.5 ml MoBiCol column (MoBiTec) and washed with 25 ml of HOPS washing buffer (HWB, 1 M NaCl, 20 mM HEPES/NaOH, pH 7.4, 1.5 mM MgCl$_2$, and 20% (v/v) glycerol). FLAG-peptide was added and incubated on a turning wheel for 40 min at 4°C. The eluate was collected by centrifugation (150×*g*, 30 s, 4°C) and concentrated in a Vivaspin 100 kDa MWCO concentrator (Sartorius), which was previously incubated for 45 min with HWB containing 1% TX-100. The concentrated eluate was applied to a Superose 6 Increase 15/150 column (Cytiva) for size exclusion chromatography (SEC) and eluted in 50 µl fractions using ÄKTA go purification system (Cytiva). Peak fractions were used for further analysis.

## Mass photometry

Mass photometry experiments were done using a Refeyn TwoMP (Refeyn Ltd). Data were obtained using AcquireMP software and analyzed using DiscoverMP (both Refeyn Ltd). Glass coverslips were used for sample analysis. Perforated silicone gaskets were placed on the coverslips to form wells for every sample to be measured. Samples were evaluated at a final concentration of 10 nM in a total volume of 20 µl in the buffer used for SEC.

## Cryo-EM sample preparation and data acquisition

Prior to cryo-EM analysis, HOPS samples were tested by negative-stain EM, using 2% (w/v) uranyl formate solution as previously described (*Januliene and Moeller, 2021*). Negative-stain micrographs were recorded manually on a JEM-2100Plus transmission electron microscope (JEOL), operating at 200 kV and equipped with a XAROSA CMOS 20 Megapixel Camera (EMSIS) at a nominal magnification of 30,000 (3.12 Å per pixel). For cryo-EM, 3 µl of 0.6–0.9 mg/ml of freshly purified wt HOPS complex were applied onto glow-discharged CF grids (R1.2/1.3) (EMS) and immediately plunge-frozen in liquid ethane using a Vitrobot Mark IV (Thermo Fisher Scientific) with the environmental chamber set to 100% humidity and 4°C. Micrographs were recorded automatically with EPU (Thermo Fisher Scientific), using a Glacios microscope (Thermo Fisher Scientific) operated at 200 kV and equipped with a Selectris energy filter and a Falcon 4 detector (both Thermo Fisher Scientific). Images were recorded in Electron Event Representation (EER) mode at a nominal magnification of 130,000 (0.924 Å per pixel) in the defocus range from −0.8 to −2.8 µm with an exposure time of 8.30 s resulting in a total electron dose of approximately 50 e⁻ Å⁻². 

## Cryo-EM image processing

All cryo-EM data processing (*Figure 1—figure supplement 1*) was performed in cryoSPARC v3.3.1 (*Punjani et al., 2017*). For all collected movies, patch motion correction (EER upsampling factor 1, EER number of fractions 40) and patch contrast transfer function estimation were performed using

cryoSPARC implementations. To solve the structure of the core part of HOPS (*Figure 1—figure supplement 1E,F* and *Figure 1—figure supplement 2*) reference-free blob particle picking on 2186 pre-processed movies of the first data set and particle extraction using a box size of 672 pixels (px, 0.924 Å per pixel) binned to 128 px was performed. Extracted particles were subjected to 2D classification to eliminate bad picks. Selected good 2D classes (representative good 2D classes are shown in *Figure 1—figure supplement 3A*) were used for template particle picking on 2186 movies from the first data set combined with additional 6580 movies from the second data set, preprocessed alike. After removal of duplicates, picked particles were extracted with the same box size and subjected to rounds of 2D classification, ab-initio reconstruction with multiple classes and 3D heterogeneous refinement to remove false positive particle picks. From the best class, particles were extracted using a box size of 672 px (0.924 Å per pixel) binned to 320 px and subjected to 2D classification. Particles from this 2D classification were also used for flexibility analysis (see below). 407,996 selected particles from good 2D classes were used for ab-initio reconstruction with six classes and followed by 3D heterogeneous refinement. This heterogeneous refinement resulted in two best classes containing 130,009 and 116,151 particles, respectively, which were further refined individually. For each of both classes, a Non-Uniform (NU)-refinement was performed, followed by particle extraction with a box size of 672 px (0.924 Å per pixel, without binning) with homogeneous and NU-refinement afterward. Resulting consensus maps were used for local refinements of different parts of the structure (*Figure 1—figure supplement 1A*, *Figure 1—figure supplement 2*, and *Figure 1—figure supplement 3B-E*). First consensus NU-refinement that reached global resolution of 4.2 Å (Fourier shell correlation [FSC]=0.143) was used for local refinements of the upper 'core' part of the complex (4 Å) and the SNARE-binding module (3.6 Å), the second consensus NU-refinement, resolved to 4.4 Å (FSC=0.143), was used for local refinements resolving bottom parts of the complex (4.4 and 5 Å).

To better resolve distal parts of the complex, flexible Vps39 and Vps41 N-terminal fragments, the following approach was used (*Figure 1—figure supplement 1B*). Here, micrographs from the first two data sets were combined with micrographs from two additional data sets of 2841 and 8338 movies. After template picking, about 3896 million particles were extracted using a box size of 882 pixels (0.924 Å per pixel) and used for rounds of heterogeneous, homogeneous, and NU-refinements to obtain 3D reconstructions, which best resolve Vps39 and Vps41. At these steps, binning was applied for particle extractions. Then, particles belonging to one of such best classes were subjected to a round of NU-refinement followed by 3D variability analyses using masks covering either Vps39 or Vps41. The further 3D variability display procedures were used to better sort particles. Finally, the best particles were extracted with the same box size (882 pixels, 0.924 Å per pixel) without binning and used for local refinements of Vps39 or Vps41 (*Figure 1—figure supplement 1B*, *Figure 1—figure supplement 3F,G*).

For all local refinements, masks were generated in UCSF Chimera (*Pettersen et al., 2004*). During processing, no symmetry was applied. The quality of final maps is demonstrated in Extended Data *Figure 3*. All FSC curves were generated in cryoSPARC. Local resolutions of locally-refined maps (Figure S3B-G) were estimated in cryoSPARC and analyzed in UCSF ChimeraX (*Pettersen et al., 2021*). Data set statistics can be found in *Supplementary file 2*.

To analyze the flexibility of the complex, a different processing scheme was applied (*Figure 1—figure supplement 1A*, dashed arrows). For this, a set of 383,881 good particles was selected after 2D classification of 320 px-binned particles (initial box size of 672 px with 0.924 Å per pixel). These particles were subjected to either ab-initio reconstruction and heterogeneous refinement with ten classes (*Figure 2—figure supplement 3A,B*) or several refinement rounds followed by 3D variability analysis (*Figure 2—figure supplement 3D*). Prior to 3D variability analysis, ab-initio reconstruction with one class, homogeneous and NU-refinement were performed.

## Model building and refinement

Models of HOPS subunits were initially generated using AlphaFold (*Jumper et al., 2021*; *Varadi et al., 2022*) and docked into locally refined maps using 'Fit in Map' tool in UCSF ChimeraX (*Pettersen et al., 2021*). The N-terminal parts of Vps41 (residues 1–863) and Vps39 (residues 1–700) and the C-terminal part of Vps16 (from residue 739) with no well-resolved densities assigned, were removed. The AlphaFold model of Vps11 was initially split into two parts ('Vps11top', residues 1–760, and 'Vps11bottom', residues 784–1025), which were first refined separately; the region predicted by

AlphaFold as unfolded (residues 761–783) was deleted. Fitting of the C-terminal parts of Vps11 (residues 784–1025) and Vps39 (residues 701–1045) was improved using Namdinator (*Kidmose et al., 2019*). Afterward, models of single proteins were manually adjusted and refined in COOT (*Emsley et al., 2010*), followed by iterative rounds of refinements against corresponding locally refined and their composite maps in Phenix (*Liebschner et al., 2019*) and COOT. In Phenix Graphical User Interface, real space refine tool (*Afonine et al., 2018*) with or without option 'rigid body' was used. After several refinement rounds, two separate models were created by joining models of the upper (Vps33, Vps16, Vps41, Vps18, and Vps11 top) and lower (Vps11 bottom, Vps39) parts of the complex. These partly combined models were subjected to further several iterations of refinements in Phenix and COOT. Afterward, the two refined models were fused into a single model, which was again refined in Phenix and COOT. Sequences of the model in the bottom part of the complex were changed to polyalanines (residues 784–1025 in Vps11, residues 1–493 in Vps18, and residues 701–1045 in Vps39), since no assignment of side chains was possible at the resolution obtained there. In other parts of the complex, where blurred densities did not allow unambiguous model building, respective short fragments of the model were also replaced by polyalanine chains or deleted. Afterward, the initially deleted part of Vps11 (residues 761–783) was built de novo according to the cryo-EM density (residues 769–783 were replaced by alanines). Finally, the complete model was subjected to another round of refinement in Phenix followed by manual refining in COOT. Model validation was performed using MolProbity (*Williams et al., 2018*). Figures were prepared using UCSF ChimeraX. Model refinement and validation statistics can be found in *Supplementary file 2*.

## ALFA pulldowns for mass spectrometry

One liter of YP medium containing 2% glucose (v/v) was inoculated with an overnight preculture. Cells were grown to $OD_{600}$ 1 at 26°C, followed by 1 hr incubation at 38°C. Cultures were harvested by centrifugation at 4800×g for 10 min at 4°C. Pellets were washed with cold Pulldown buffer (PB), 150 mM KAc, 20 mM HEPES/NaOH, pH 7.4, 5% (v/v) glycerol, and 25 mM CHAPS. The pellet was resuspended in a 1:1 ratio (w/v) in PB. supplemented with Complete Protease Inhibitor Cocktail (Roche) and afterward dropwise frozen in liquid nitrogen before lysed in 6875D LARGE FREEZER/MILL (SPEX SamplePrep LLC). Powder was thawed on ice and resuspended in PB by using a glass pipette, followed by two centrifugation steps at 5000 and 15,000×g at 4°C for 10 and 20 min. The supernatant was added to 12.5 µl prewashed ALFA Selector ST beads (2500×g, 2 min, 4°C) (NanoTag Biotechnologies) and incubated for 15 min at 4°C while rotating on a turning wheel. After incubation, beads were washed two times in PB and four times in PB without CHAPS. Samples were digested using PreOmics sample kit (iST Kit, preomics) and analyzed in Q ExativePlus mass spectrometer (Thermo Fisher Scientific).

## GST pulldowns

Nucleotide-specific interaction of the Rab-GTPase Ypt7 with purified HOPS variants was analyzed in GST pulldowns using GST-Ypt7 or GST-Ypt1 as a negative control. About 125 µg purified Rab-GTPases was preloaded with 1 mM GDP or GTP in the presence of 20 mM EDTA and wash buffer (150 mM NaCl, 50 mM HEPES-NaOH, pH 7.4, 2 mM $MgCl_2$, 0.1 % Triton X-100) in a water bath for 30 min at 30°C. For nucleotide stabilization, 25 mM $MgCl_2$ was added. Prewashed GSH Sepharose 4B (GE Healthcare) was added to loaded Rab-GTPases and incubated for 1 hr at 4°C on a turning wheel. Beads were centrifuged for 1 min at 300×g before adding 25 µg of respective HOPS variants, followed by 1.5 hr incubation at 4°C on a turning wheel. Beads were washed three times with wash buffer, followed by two elution steps with 600 µl wash buffer containing 20 mM EDTA. After an incubation at room temperature for 20 min while rotating, the supernatant was TCA precipitated. About 10% of the final sample was loaded on a 7.5% SDS gel next to 5% of protein input. Samples were analyzed via western blot using antibodies against the FLAG-Tag. Beads were boiled in 50 µl Laemmli buffer. About 2% of the sample was loaded on an 11% SDS gel for Coomassie staining as Rab-GTPase loading control. Bands were quantified relative to the Rab-GTPase content.

## Tethering assay

HOPS-mediated tethering assays were performed as described (*Füllbrunn et al., 2021*). For this, ATTO488 labeled liposomes were prepared and loaded with prenylated Ypt7 (*Langemeyer et al.,*

*2018*). About 50 nmole liposomes was incubated with 50 pmole pYpt7:GDI complex together with GTP for 30 min at 27°C. For reactions, 0.17 mM Ypt7-loaded liposomes were incubated with different concentrations of HOPS complex or buffer (300 mM NaCl, 20 mM HEPES-NaOH, pH 7.4, 1.5 mM $MgCl_2$, 10 % (v/v) glycerol) for 10 min at 27°C, followed by sedimentation for 5 min at 1000×*g*. The overall tethered liposomes in the pellet fraction were determined in a SpectraMax M3 fluorescence plate reader (Molecular Devices) comparing the ATTO488 fluorescent signal of the supernatant before and after sedimentation.

## Fusion assays

Fusion assays and the purification of all proteins were performed as described (*Langemeyer et al., 2018*) with a protein to lipid ratio of 1:8000. Reconstituted proteoliposomes (RPLs) were composed of the vacuole mimicking lipid (VML) mix (*Zick and Wickner, 2014*). One population of RPLs carried the SNARE Nyv1 and the other set contained Vti1 and Vam3. RPLs were preloaded with prenylated Ypt7 with the help of 100 mM Mon1-Ccz1 and 0.5 mM GTP (*Langemeyer et al., 2018*). Then, 25 nM HOPS complex, 50 nM Sec18, 600 nM Sec17, and finally 100 nM Vam7 were added. Fusion of liposomes was followed by content mixing of the RPLs and subsequent increase in fluorescence which was monitored in a SpectraMax M3 fluorescence plate reader (Molecular Devices).

## Cloning and protein purification of CtVPS391-500

Codon-optimized synthetic DNA (GenScript) encoding amino acids 1–500 of Vps39 from *C. thermophilum* (CtVp391-500; NCBI XP_006691033) was subcloned into a modified pET28a expression vector yielding an N-terminally His6-SUMO-tagged fusion protein (His6-SUMO-CtVps391-500). In short, His6-SUMO-CtVps39 was purified by Ni-NTA affinity chromatography followed by proteolytic cleavage with SUMO protease at 4°C overnight. SEC was performed to separate CtVps391-500 from the expression tag and SUMO protease and yielded >95% pure protein. To obtain phase information for structure determination, selenomethionine-substituted CtVps391-500 was prepared according to well-established methods (*Doublié, 1997*).

## Limited proteolysis and protein crystallization

Since initial crystallization approaches did not yield protein crystals suitable for X-ray structure determination, flexible parts of CtVps391-500 were removed by limited proteolysis with α-Chymotrypsin (Merck) at 37°C for 10 min. Limited proteolysis was stopped by addition of protease inhibitors (Pierce Protease Inhibitor Tablets, EDTA-free, Thermo Fisher Scientific), and an additional SEC in buffer containing 20 mM BIS-TRIS pH 6.5, 200 mM NaCl, and protease inhibitors (1:1000) was performed as a final polishing step. Best crystals were obtained by seeding at 12°C and a protein concentration of 7.5 mg/ml in a crystallization condition containing 0.1 M MES pH 7.25 and 20% PEG 2000 MME. Selenium-derivative crystals were flash-cooled in liquid nitrogen in latter condition with 25% glycerol as cryoprotectant.

## Crystal structure determination

Anomalous X-ray data were collected from a single crystal at 100K at beamline P13, EMBL Hamburg, Germany and diffraction data were processed using XDSAPP3 (*Krug et al., 2012*). Phase determination using single-wavelength anomalous dispersion at the selenium peak was not successful. A manually trimmed AlphaFold (*Jumper et al., 2021*) model of CtVps39 1-500 was used as a search template in molecular replacement and yielded a single solution with a TFZ score of 24.8 in phenix.phaser (*Adams et al., 2010*; *Liebschner et al., 2019*; *McCoy et al., 2007*), which was then used for MR-SAD phasing in phenix.autosol (*Adams et al., 2010*; *Liebschner et al., 2019*; *Terwilliger et al., 2009*) and subsequent density modification and automated model building using phenix.autobuild (*Adams et al., 2010*; *Liebschner et al., 2019*; *Terwilliger et al., 2009*). Iterative cycles of model building in COOT (*Emsley et al., 2010*) and refinement using phenix.refine (*Afonine et al., 2012*; *Liebschner et al., 2019*; *Terwilliger et al., 2009*) led to a final model of CtVps391-500 with Rfactors of Rwork 26.9% and Rfree 29.8%. The model contained no Ramachandran outliers with 96.74% residues within favored regions. Crystallographic statistics are summarized in *Supplementary file 3*. PYMOL and UCSF ChimeraX were used for visualization and graphical analysis.

## Acknowledgements

The authors thank Kristian Parey for help with model building, and all members of the Ungermann and Moeller lab for feedback. This work was supported by the SFB 944 (P11 to CU; P16 to DK, P20 to FF, P26 to AM), the DFG (UN111/5-6 to CU and MO 2752/3-6 to AM), the DFG INST190/196-1 FUGG (AM) and the BMBF/DLR 01ED2010 (AM).

## Additional information

### Funding

| Funder | Grant reference number | Author |
|---|---|---|
| Deutsche Forschungsgemeinschaft | UN111/5-6 | Christian Ungermann |
| Deutsche Forschungsgemeinschaft | INST190/196-1 FUGG | Arne Moeller |
| Bundesministerium für Bildung und Forschung | BMBF/DLR 01ED2010 | Arne Moeller |
| Deutsche Forschungsgemeinschaft | SFB 944 P11 | Christian Ungermann |
| Deutsche Forschungsgemeinschaft | SFB 944 P27 | Arne Moeller |
| Deutsche Forschungsgemeinschaft | SFB 944 P20 | Florian Fröhlich |
| Deutsche Forschungsgemeinschaft | SFB 944 P16 | Daniel Kümmel |
| Deutsche Forschungsgemeinschaft | MO 2752/3-6 | Arne Moeller |

The funders had no role in study design, data collection and interpretation, or the decision to submit the work for publication.

### Author contributions

Dmitry Shvarev, Conceptualization, Data curation, Formal analysis, Validation, Investigation, Visualization, Methodology, Writing - original draft, Writing - review and editing; Jannis Schoppe, Conceptualization, Formal analysis, Validation, Investigation, Visualization, Methodology, Writing - review and editing; Caroline König, Conceptualization, Data curation, Formal analysis, Validation, Investigation, Visualization, Methodology, Writing - review and editing; Angela Perz, Nadia Füllbrunn, Dovile Januliene, Data curation, Formal analysis, Validation, Investigation, Methodology, Writing - review and editing; Stephan Kiontke, Data curation, Formal analysis, Validation, Investigation, Visualization, Methodology, Writing - review and editing; Lars Langemeyer, Data curation, Supervision, Validation, Investigation, Methodology, Writing - review and editing; Kilian Schnelle, Data curation, Software, Formal analysis, Validation, Investigation, Visualization, Methodology; Daniel Kümmel, Data curation, Formal analysis, Supervision, Funding acquisition, Validation, Visualization, Methodology, Writing - review and editing; Florian Fröhlich, Data curation, Formal analysis, Funding acquisition, Validation, Visualization, Methodology, Writing - review and editing; Arne Moeller, Conceptualization, Data curation, Formal analysis, Supervision, Funding acquisition, Validation, Visualization, Methodology, Writing - original draft, Project administration; Christian Ungermann, Conceptualization, Supervision, Funding acquisition, Validation, Writing - original draft, Project administration, Writing - review and editing

### Author ORCIDs

Dmitry Shvarev (ID) http://orcid.org/0000-0002-9776-268X
Stephan Kiontke (ID) http://orcid.org/0000-0001-5822-913X
Lars Langemeyer (ID) http://orcid.org/0000-0002-4309-0910
Dovile Januliene (ID) http://orcid.org/0000-0002-3279-7590
Kilian Schnelle (ID) http://orcid.org/0000-0001-8808-594X

Daniel Kümmel  http://orcid.org/0000-0003-3950-5914
Florian Fröhlich  http://orcid.org/0000-0001-8307-2189
Arne Moeller  http://orcid.org/0000-0003-1101-5366
Christian Ungermann  http://orcid.org/0000-0003-4331-8695

**Decision letter and Author response**
Decision letter https://doi.org/10.7554/eLife.80901.sa1
Author response https://doi.org/10.7554/eLife.80901.sa2

## Additional files

### Supplementary files
- Supplementary file 1. Yeast strains used in the study.
- Supplementary file 2. Cryo-EM data collection, refinement and validation statistics.
- Supplementary file 3. Crystallographic data collection and refinement statistics (molecular replacement).
- MDAR checklist

### Data availability
All diffraction data are deposited in the PDB as indicated in the manuscript. PDB files are mentioned there.

The following datasets were generated:

| Author(s) | Year | Dataset title | Dataset URL | Database and Identifier |
|---|---|---|---|---|
| Shvarev D, Schoppe J, Konig C, Perz A, Fullbrunn N, Kiontke S, Langemeyer L, Januliene D, Schnelle K, Kummel D, Frohlich F, Moeller A, Ungermann C | 2022 | HOPS tethering complex from yeast | https://www.emdataresource.org/EMD-14964 | EMDataResource, EMD-14964 |
| Shvarev D, Schoppe J, Koenig C, Perz A, Fuellbrunn N, Kiontke S, Langemeyer L, Januliene D, Schnelle K, Kuemmel D, Froehlich F, Moeller A, Ungermann C | 2022 | HOPS tethering complex from yeast, consensus map covering the upper part of the complex | https://www.emdataresource.org/EMD-14965 | EMDataResource, EMD-14965 |
| Shvarev D, Schoppe J, Koenig C, Perz A, Fuellbrunn N, Kiontke S, Langemeyer L, Januliene D, Schnelle K, Kuemmel D, Froehlich F, Moeller A, Ungermann C | 2022 | HOPS tethering complex from yeast, consensus map covering the bottom part of the complex | https://www.emdataresource.org/EMD-14966 | EMDataResource, EMD-14966 |
| Shvarev D, Schoppe J, Koenig C, Perz A, Fuellbrunn N, Kiontke S, Langemeyer L, Januliene D, Schnelle K, Kuemmel D, Froehlich F, Moeller A, Ungermann C | 2022 | HOPS tethering complex from yeast, local refinement map of the SNARE-binding module | https://www.emdataresource.org/EMD-14967 | EMDataResource, EMD-14967 |
| Kiontke S, Ungermann C, Kuemmel D | 2022 | Structure of Vps39 N-terminal domain from Chaetomium thermophilum | https://www.rcsb.org/structure/7ZTY | RCSB Protein Data Bank, 7ZTY |

*Continued on next page*

*Continued*

| Author(s) | Year | Dataset title | Dataset URL | Database and Identifier |
|---|---|---|---|---|
| Shvarev D, Schoppe J, Koenig C, Perz A, Fuellbrunn N, Kiontke S, Langemeyer L, Januliene D, Schnelle K, Kuemmel D, Froehlich F, Moeller A, Ungermann C | 2022 | HOPS tethering complex from yeast | https://www.rcsb.org/structure/7ZU0 | RCSB Protein Data Bank, 7ZU0 |
| Shvarev D, Schoppe J, Koenig C, Perz A, Fuellbrunn N, Kiontke S, Langemeyer L, Januliene D, Schnelle K, Kuemmel D, Froehlich F, Moeller A, Ungermann C | 2022 | HOPS tethering complex from yeast, local refinement map of the backbone part of the complex | https://www.emdataresource.org/EMD-14968 | EMDataResource, EMD-14968 |
| Shvarev D, Schoppe J, Koenig C, Perz A, Fuellbrunn N, Kiontke S, Langemeyer L, Januliene D, Schnelle K, Kuemmel D, Froehlich F, Moeller A, Ungermann C | 2022 | HOPS tethering complex from yeast, local refinement map of the bottom part of the complex (Vps18) | https://www.emdataresource.org/EMD-14969 | EMDataResource, EMD-14969 |
| Shvarev D, Koenig C, Perz A, Fuellbrunn N, Kiontke S, Langemeyer L, Januliene D, Schnelle K, Kuemmel D, Froehlich F, Moeller A, Ungermann C | 2022 | HOPS tethering complex from yeast, local refinement map of the bottom part of the complex (Vps39) | https://www.emdataresource.org/EMD-14970 | EMDataResource, EMD-14970 |

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
