## [Editor Report]

This landmark study reports the cryo-EM structure of HOPS, a heterohexameric tether that participates in the fusion of late endosomes, autophagosomes, and AP-3 vesicles with lysosomes. The structure provides a convincing update of earlier, lower-resolution models. Interestingly, the SNARE-binding module is attached to the core of the complex. These results suggest possible mechanisms by which HOPS could catalyze SNARE-dependent fusion.

---

## [Decision Letter]

**Decision letter after peer review:**

Thank you for submitting your article "Structure of the lysosomal membrane fusion machinery" for consideration by *eLife*. Your article has been reviewed by 3 peer reviewers, including Benjamin S Glick as Reviewing Editor and Reviewer #1, and the evaluation has been overseen by Vivek Malhotra as the Senior Editor.

All of the reviewers are excited about the novel (and beautiful) structure of HOPS. This advance is a major step forward for the field, and it is especially important because HOPS has been studied so intensively. However, the reviewers are also unanimous in expressing concern about the functional interpretations. The conclusion that HOPS has a rigid core is not adequately justified. This uncertainty, combined with the wobbly membrane connections at the ends of the complex, cast doubt on the idea that HOPS acts as a stiff scaffold to promote fusion. In addition, Reviewer #2 points out that Vps33 is unlikely to remain engaged with the SNAREs as they zipper. Unless the authors can bolster their arguments far more effectively, the manuscript needs to be thoroughly rewritten to remove the unsupported claims.

Instead, the authors are encouraged to mine the structural data for further insights based on novel aspects of the structure, comparisons with other multi-subunit tethers, and fresh interpretations of the extensive literature on HOPS. This manuscript has the potential to be an elegant contribution if the structure is placed in a more appropriate context.

These suggestions are fleshed out in the detailed comments provided by the reviewers.

*Reviewer #1 (Recommendations for the authors):*

The data are well presented, and I have no detailed recommendations.

*Reviewer #2 (Recommendations for the authors):*

I suspect that the structure itself, with extensively intertwined subunits, is also novel. Overall, the paper is a must-accept for *eLife*, yet the following points also must be addressed, necessitating substantial re-writing but not requiring additional experiments.

Some questions about the paper and its presentation:

1. Repeated claims are made about which parts of the structure are flexible and which are rigid, yet there are no measurements of structural bending vs force applied ie, rigidity. If parts of the structure showed little variability among the many images analyzed, I presume that this was taken as a measure of rigidity; is there literature to support this, and if so it should be cited and discussed. How do the imputed bending energies compare to the binding energies for the things which bind to HOPS, such as Ypt7 and the SNAREs?

2. The model appears to claim that Vps33 would bind Nyv1 and Vam3 and then Vps33 will continue to be engaged with the SNAREs as the SNAREs zipper. What is the basis for this claim? Conventional thought is that each of these SNAREs is bound to a site on Vps33 with their R or Q residue and the apolar heptad repeat residues facing "in" to the binding site, in parallel and in the register as presented in 2015 by Baker et al., but that for zippering to proceed these side chains will have to leave these sites and face each other in the center of the 4-helical 4-SNARE coiled coils.

The structure of HOPS certainly is in accord with its well-established function as a tether and shows that Vps33 is off to the side between the two Ypt7 binding sites, Vps39 and Vps41, well-positioned to engage SNAREs to perform their alignment. It's unclear how the structure goes beyond that functionally. This is not to diminish the beauty and novelty of the structure, but to question their repeated claims that the structure now reveals substantial new insights about function.

3. Figures 1 and 2 are models of clarity and elegance.

4. With all respect for the effort that went into the Figure 3 experiment, it adds little to the paper or to mechanistic understanding. Unsurprisingly, the deletion in Vps41 blocks both tethering and fusion, while the deletions in Vps11 and 18 block fusion but not tethering. Figure 3 simply suggests that there's more to HOPS than a rigid structure and 3 active sites, one with Sec1/Munc18 function and two for tethering by means of binding Ypt7.

5. Readers will want to know whether Vps39 and 41 structures are similar to structures of other proteins that bind stoichiometrically to Rabs. Are there other proteins with a similar overall structure as the HOPS structure presented here, or is this extended structure with extensively interdigitated chains unique? Substantial additional discussion to place this structure in the context of others would build on the core of the paper, the determination of HOPS structure.

6. An earlier structure of crosslinked HOPS from this group proposed that each of the 6 subunits had an overall globular fold and that these subunits had binding surfaces adjacent to each other. In contrast, the current structure shows that there is extensive interdigitation of the subunits. Is this seen for many other proteins?

7. The claim that "HOPS complexes lacking β-propellers in Vps11 or Vps18 showed an alteration in the relative orientation of Vps39 within the complex, which indicates a more flexible backbone, indicating that backbone integrity is essential for full HOPS activity" (lines 207 to 209) is particularly puzzling. There are no data about whether these deletions are more or less flexible, and it's unclear how this can be inferred from an altered Vps39 orientation. There are no data showing that the loss of HOPS activity is caused by loss of rigidity.

The paper should be re-written to remove claims that it reveals new aspects of function or of rigidity but should emphasize that this is a novel structure.

*Reviewer #3 (Recommendations for the authors):*

With regard to the structural analyses, it would be useful to speculate on the possible cause(s) for the difference between this structure and previous lower resolution structures, perhaps illuminating some additional (perhaps biologically relevant?) differences? Is the rigid structure shown in this manuscript the only one possible for HOPS? Or are there additional conformations accessible in vivo that are not represented with this purified cryoEM sample? Does the fact that HOPS was massively overexpressed for purification indicate that certain post-translational modifications may not be present?

Are the resolution and map really suitable to accurately model the RING finger domain residues (Figure 2 and Figure 2-S1), or to map disease-associated mutations (Figure 4-S1)?

In Figure 3-S1 and the text, how good is the evidence that these Ypt7 binding sites predicted by Alphafold are accurate, especially as the location does not make a lot of biological sense with the location of the ALPS motif and membrane binding of Ypt7?

The authors' focus on the "rigid core" of HOPS, with respect to how this could help facilitate SNARE-mediated fusion, is a bit unclear, especially with the noted flexibility of both the Vps39 and Vps41 ends. Also, it is unclear how to interpret the angle differences of these flexible regions, in terms of overall length changes and possible changes during the tethering and fusion reactions.

With regard to this rigid core, the claim is that a number of helices "are tightly interlocked through coiled-coil motifs" (Line 127, Page 8), but the images in Figure 2 do not appear to be particularly interlocked or tight. Perhaps a calculation of buried surface might support this description?

Interpretation of the negative stain images in Figure 3D (especially with vps18-deltaN) is not entirely convincing, especially as the gel in Figure 1-S4 seems to indicate that Vps41 is missing from the vps18-deltaN pulldown? Were these mutants analyzed by mass spectrometry (similar to that for the RING finger mutants in Figure 2-S2?)? It would also be helpful to speculate more mechanistically about why these complexes can tether, but not drive fusion.

[Editors’ note: further revisions were suggested prior to acceptance, as described below.]

Thank you for resubmitting your work entitled "Structure of the HOPS tethering complex, a lysosomal membrane fusion machinery" for further consideration by *eLife*. Your revised article has been evaluated by Vivek Malhotra (Senior Editor) and a Reviewing Editor.

The manuscript has been improved but the reviewers are not yet fully satisfied with the changes. They ask that claims of rigidity either be backed up with solid evidence or else presented as untested speculation, and that claims that this study tells us about how HOPS works be removed from earlier parts of the manuscript and moved to the Discussion as a model.

Below are the comments of the three reviewers. These issues need to be addressed for publication in *eLife*.*Reviewer #1 (Recommendations for the authors):*

– While the authors are free to propose their favored interpretations, you need to separate speculation from evidence-based interpretation.

– Rigidity has a well-defined meaning as resistance to deformation induced by stress. Nothing in this study measures rigidity. The central core of HOPS may be conformationally stable to thermal fluctuations as an isolated complex, but there is no way to predict how resistant it would be to deformation when stressed by a process such as SNARE zippering.

– The authors propose that Vps33 lets go of the SNAREs during zippering but that due to other interactions, HOPS remains tightly enough associated with the SNAREs to promote fusion. That idea might be correct but no experimental support is provided, so it is a weak link in the model.

– Another weak link is that the flexibility of the Ypt7-binding ends of HOPS are still challenging to reconcile with the idea that HOPS is a rigid scaffold.

– The bottom line is that as before, this study represents a major advance by describing a high-quality structure of HOPS. But the authors need to take seriously our request to separate speculation from interpretation, and to tone down the conclusion that this structure has revealed a new mechanism.

*Reviewer #2 (Recommendations for the authors):*

The authors clearly thought that his manuscript only needed a few minor tweaks, and didn't do a thorough re-write or come to grips with our central points.

Ungermann and his colleagues are still confusing protein stability with rigidity. Well-resolved parts of structures must be stable, but may or may not be rigid, the former term referring to the energy needed to unfold, either locally or globally, the latter to the energy needed to deform conformation. The data give no indication on rigidity, while the authors strongly leans on rigidity in speculations about HOPS function. Ungermann hasn't addressed this at all, and the two references he cites in his reviewers responses are neither cited nor discussed in the revised manuscript.

The Abstract remains sadly misleading, stating that "HOPS is surprisingly rigid" and "The SNARE binding module is rigidly attached to the core.." (see above). The last sentence, "Our results explain HOPS dual functionality and unravel why tethering complexes are …essential…fusion" is therefore very inaccurate.

There are no data on rigidity, as opposed to which regions of HOPS fold stably, but the manuscript persists in this vein. At the bottom of p6, they state "..it was unknown how tethering and fusion activities of HOPS are linked mechanistically. To clarify this…" but what follows addresses what a tethering structure of HOPS binding its Rab might look like through predictions and modeling. This doesn't address how "tethering and fusion activities of HOPS are linked mechanistically".

They show that removing the β-propeller of Vps41 alters HOPS structure, but confuse this with an altered rigidity.

The last paragraph before the Discussion is speculation and should be explicitly labeled as such and put in the Discussion. They present no evidence of HOPS continued association with the SNAREs as zippering proceeds. Their speculative model wasn't altered at all, and still shows Vps33 associating with the SNAREs as zippering progresses, though he acknowledges that this is very unlikely.

In short, the authors didn't take our reviews seriously. Re-reading the decision letter now, it was very clear that we wanted a thorough re-write, but we didn't get it.

I suggest that we return the paper to the authors, and that they take our reviews to heart and thoroughly re-write this paper.

*Reviewer #3 (Recommendations for the authors):*

Overall, I am still excited about this structure, and what it suggests for how tethering complexes can function. However, although some good changes have been made to the revised version, some of the conclusions are more speculative than the text would suggest.

Some of the changes in the revision still do not adequately address all the reviewers' concerns. Most importantly, the idea of the central core being "rigid" is clearly supported by lack of other conformations in the cryoEM data, but that conclusion is still speculative with regard to the conformation during the fusion reaction in vivo, and needs to be softened a bit in the text. The fact that other conformations are not observed in the structure indicates that the purified complex under these conditions does not seem flexible. However, the structure may just represent the lowest energy, stable state. The data does not indicate that this region is "rigid" in a way that suggests that it cannot bend under other conditions, e.g. in vivo. Certainly future experiments will help understand this, but the readers should not come away with this idea as definitive.

Although a little more information regarding other tethering complexes/membrane fusion machinery was added, it still is likely to be more relevant for a specialized readership, not a broad *eLife* audience.

This previous concern was not addressed: "With regard to the structural analyses, it would be useful to speculate on the possible cause(s) for the difference between this structure and previous lower resolution structures, perhaps illuminating some additional (perhaps biologically relevant?) differences?" It is unclear how improvements and acceleration in the sample preparation helped yield a "rigid" molecule vs. previous studies.

"In Figure 3-S1 and the text, how good is the evidence that these Ypt7 binding sites predicted by Alphafold are accurate, especially as the location does not make a lot of biological sense with the location of the ALPS motif and membrane binding of Ypt7?" For this concern, I respectfully disagree that all Alphafold predictions are that accurate, especially if the biological conclusion is unclear, and this speculation should be better explained or the conclusion softened.

---

## [Author Response]

Reviewer #1 (Recommendations for the authors):The data are well presented, and I have no detailed recommendations.

We thank the Reviewer for her/his kind comment.

Reviewer #2 (Recommendations for the authors):I suspect that the structure itself, with extensively intertwined subunits, is also novel. Overall, the paper is a must-accept for eLife, yet the following points also must be addressed, necessitating substantial re-writing but not requiring additional experiments.

We thank the Reviewer for the kind words and helpful comments. Below and in our revised manuscript, we have addressed the concerns raised.

Some questions about the paper and its presentation:1. Repeated claims are made about which parts of the structure are flexible and which are rigid, yet there are no measurements of structural bending vs force applied ie, rigidity. If parts of the structure showed little variability among the many images analyzed, I presume that this was taken as a measure of rigidity; is there literature to support this, and if so it should be cited and discussed. How do the imputed bending energies compare to the binding energies for the things which bind to HOPS, such as Ypt7 and the SNAREs?

Averaging of individual projection views is fundamental to cryo-EM. Therefore, during image processing, rigid particles (or their rigid parts) reach higher resolutions than flexible ones. According to our extensive 2D and 3D classification and variability analyses, we conclude that the core part of HOPS is the most rigid part of the complex, since it shows the highest resolution and better 2D-3D alignment (the general problem of internal particle flexibility in cryo-EM is also addressed in the literature, for example in Rawson et al. *Methods* 2016 and Punjani and Fleet *J. Struct. Biol.* 2021).

The affinities between HOPS and Ypt7 are relatively low (in the μM range); given the rigidity of the complex with its large hydrophobic interface (see reply to reviewer #1), the binding energies will therefore be minor.

2. The model appears to claim that Vps33 would bind Nyv1 and Vam3 and then Vps33 will continue to be engaged with the SNAREs as the SNAREs zipper. What is the basis for this claim? Conventional thought is that each of these SNAREs is bound to a site on Vps33 with their R or Q residue and the apolar heptad repeat residues facing "in" to the binding site, in parallel and in the register as presented in 2015 by Baker et al., but that for zippering to proceed these side chains will have to leave these sites and face each other in the center of the 4-helical 4-SNARE coiled coils.The structure of HOPS certainly is in accord with its well-established function as a tether and shows that Vps33 is off to the side between the two Ypt7 binding sites, Vps39 and Vps41, well-positioned to engage SNAREs to perform their alignment. It's unclear how the structure goes beyond that functionally. This is not to diminish the beauty and novelty of the structure, but to question their repeated claims that the structure now reveals substantial new insights about function.3. Figures 1 and 2 are models of clarity and elegance.

We thank the Reviewer for her/his nice words.

4. With all respect for the effort that went into the Figure 3 experiment, it adds little to the paper or to mechanistic understanding. Unsurprisingly, the deletion in Vps41 blocks both tethering and fusion, while the deletions in Vps11 and 18 block fusion but not tethering. Figure 3 simply suggests that there's more to HOPS than a rigid structure and 3 active sites, one with Sec1/Munc18 function and two for tethering by means of binding Ypt7.

We respectfully disagree with the reviewer, as Figure 3 adds important experimental perspectives. For example, we were surprised that the deletion of the b-propeller domains of Vps11 and Vps18 resulted in a complex with the same tethering ability like the wild-type HOPS, but with almost complete inactivity with respect to fusion. As Vps39 changes its relative orientation in the 2D classes of the mutants in comparison to wild-type HOPS, we conclude that the particle gains flexibility. In our view, the mutant HOPS has lower specific activity to promote fusion. We agree that there is more to this, but our observation is very similar to findings of Baker et al., 2015. Here, a Vps33 mutation resulted in a HOPS complex that could promote fusion of liposomes with a high SNARE density like wild-type. However, at low SNARE density, this complex was inactive, whereas the wild-type HOPS promoted fusion. We find it important to correlate structure to function and, therefore, consider Figure 3 an essential part of our study.

5. Readers will want to know whether Vps39 and 41 structures are similar to structures of other proteins that bind stoichiometrically to Rabs. Are there other proteins with a similar overall structure as the HOPS structure presented here, or is this extended structure with extensively interdigitated chains unique? Substantial additional discussion to place this structure in the context of others would build on the core of the paper, the determination of HOPS structure.

To our knowledge, such an extended structure that includes intertwined α solenoids, coiled coils and RING finger domains which together serve to locate functional modules (β-propellers) to the periphery is quite unique. For instance, the exocyst complex that mediates the tethering of post-Golgi secretory vesicles to the plasma membrane has a completely different structure with α-solenoids and long α-helices (Mei et al. *Nat. Struct. Mol. Biol.* 2018). EEA1 or similar long coiled-coil tethers function completely different. They do not bind SM proteins, nor do they bind SNAREs. Their binding to Rabs is thus not comparable. We followed the reviewer’s suggestion and discuss these aspects now in more detail in the manuscript.

6. An earlier structure of crosslinked HOPS from this group proposed that each of the 6 subunits had an overall globular fold and that these subunits had binding surfaces adjacent to each other. In contrast, the current structure shows that there is extensive interdigitation of the subunits. Is this seen for many other proteins?

The previous structure of HOPS was obtained using the method of negative-stain TEM, which can only provide low resolution enabling only domain-based interpretations which necessarily lead to uncertainties during the interpretation. Our current results are improved dramatically and demonstrate the most detailed structure of HOPS to date. Consequently, we were only now able to accurately access the molecular structure.

7. The claim that "HOPS complexes lacking β-propellers in Vps11 or Vps18 showed an alteration in the relative orientation of Vps39 within the complex, which indicates a more flexible backbone, indicating that backbone integrity is essential for full HOPS activity" (lines 207 to 209) is particularly puzzling. There are no data about whether these deletions are more or less flexible, and it's unclear how this can be inferred from an altered Vps39 orientation. There are no data showing that the loss of HOPS activity is caused by loss of rigidity.The paper should be re-written to remove claims that it reveals new aspects of function or of rigidity but should emphasize that this is a novel structure.

We suggest that altering of Vps39 orientation in the mutants can be an indication of increased flexibility of the complex. We agree with the Reviewer that more data is needed in order to confirm the importance of backbone rigidity for HOPS functioning. We have revised the manuscript (lines 209-211) with words “which might be a result of increased flexibility of mutant HOPS due to the lack of structural support by β-propellers of Vps18 and Vps11, indicating that wild-type HOPS rigidity is essential for its full activity”. Currently, we are preparing additional mutants to probe the rigidity of the complex, which will be in the scope of another manuscript in the future.

Reviewer #3 (Recommendations for the authors):With regard to the structural analyses, it would be useful to speculate on the possible cause(s) for the difference between this structure and previous lower resolution structures, perhaps illuminating some additional (perhaps biologically relevant?) differences? Is the rigid structure shown in this manuscript the only one possible for HOPS? Or are there additional conformations accessible in vivo that are not represented with this purified cryoEM sample? Does the fact that HOPS was massively overexpressed for purification indicate that certain post-translational modifications may not be present?

We cannot exclude that other conformations or modifications of HOPS exist in vivo. We also agree with the Reviewer that the solved structure might allow for other conformations in vivo, possibly due to binding partners such as Rabs. Our preliminary data did not reveal that addition of Ypt7 in very high concentrations altered HOPS structure. However, further studies involving additional methods like cryo-ET or analyzing HOPS in the membrane context will be necessary to get more insight into the structure and mechanism of HOPS in vivo.

Are the resolution and map really suitable to accurately model the RING finger domain residues (Figure 2 and Figure 2-S1), or to map disease-associated mutations (Figure 4-S1)?

We agree with the Reviewer that the resolution in the region of Vps11 RING finger domain is not high enough to assign the residues unambiguously, hence we refrained from showing any side chains in the figure. However, the resolution in the upper part of the complex is sufficient for accurate residue assignment. In Figure 2 and Figure 2-S1 we attempted to highlight the general architectural similarity between Vps18 and Vps11 modular elements composed of a long helix followed by a RING finger domain and did not to focus on specific residues of the RING finger domains. In Figure 2-S1, where we additionally compared the structures of these modular elements, we, due to limitations caused by the resolution of our map, assigned the residues not only based on our structure but also with the help of respective AlphaFold models which we used for model building.

In Figure 4-S1, we aimed to show that most of the mutations are generally located at the interfaces between the subunits or in the elements important for subunit interactions without going into detail about the specific roles of residues affected by the mutations. We therefore think that the resolution of our structure is sufficient for this purpose.

In Figure 3-S1 and the text, how good is the evidence that these Ypt7 binding sites predicted by Alphafold are accurate, especially as the location does not make a lot of biological sense with the location of the ALPS motif and membrane binding of Ypt7?

Since AlphaFold has already demonstrated high quality of structure prediction of protein complexes (Evans et al. *bioRxiv* 2022), we have little reasons to not trust the accuracy of AlphaFold prediction. However, we are currently conducting mutational analyses and are planning further structural studies of HOPS and its partners in order to confirm or correct AlphaFold predictions of Ypt7-interaction sites.

The authors' focus on the "rigid core" of HOPS, with respect to how this could help facilitate SNARE-mediated fusion, is a bit unclear, especially with the noted flexibility of both the Vps39 and Vps41 ends. Also, it is unclear how to interpret the angle differences of these flexible regions, in terms of overall length changes and possible changes during the tethering and fusion reactions.

In our model, the flexibility of HOPS is limited to the distal termini of the complex, which might assist Ypt7 locating binding by HOPS and membrane tethering. However, when HOPS is already connected to both membranes, we assume it to be a much more rigid particle due to fixation of Vps41 and Vps39 at the membranes and because of the stiffness of HOPS core. The flexibility of Vps41 and Vps39 is in the range of 10-20°, which is not much. Moreover, there is evidence that the binding site to Ypt7 faces away from the membrane, thus limiting also the contribution of their hypervariable domains. This limited flexibility will position the stiff core at the membrane interface, when bound to SNAREs. Consequently, such a positioning would provide stability to the SNARE-binding module and thereby facilitate fusion.

With regard to this rigid core, the claim is that a number of helices "are tightly interlocked through coiled-coil motifs" (Line 127, Page 8), but the images in Figure 2 do not appear to be particularly interlocked or tight. Perhaps a calculation of buried surface might support this description?

The rigidity of the core is provided by the large area of interface between Vps11 and Vps18 alphasolenoids, which is 1972 A^2^ according to the PDBePISA tool. We have also added additional Supplement Figure to Figure 2 (Figure 2-S4) describing the interaction between Vps11 and Vps18 and revised the manuscript text in line 111 with words “…large interface area of 1972 Å2 which provides a…”. The intertwined long helices are rather essential for tight coupling of functional modules (Vps41, Vps39, SNARE-binding module) to the core of HOPS but not for the interaction between the core subunits Vps11/18.

Interpretation of the negative stain images in Figure 3D (especially with vps18-deltaN) is not entirely convincing, especially as the gel in Figure 1-S4 seems to indicate that Vps41 is missing from the vps18-deltaN pulldown? Were these mutants analyzed by mass spectrometry (similar to that for the RING finger mutants in Figure 2-S2?)? It would also be helpful to speculate more mechanistically about why these complexes can tether, but not drive fusion.

We suggest that due to the reduced rigidity of the complex, the mutants would not be able to fix the SNARE-module in place for effective SNARE-mediated fusion. For tethering rigidity is not as essential, since it is mediated mostly by the already quite flexible Vps41 and Vps39 subunits.

We made no observations that the mutant complex (HOPS Vps18-deltaN) is not stable. We also performed Cryo EM analysis where we could clearly see a full assembled complex. It contains Vps33 in the same stoichiometry like wild-type HOPS. Regarding Figure 1 S4, it looks like that there is one subunit missing due to tagging of the complex. In the mutants (Vps11-deltaN and Vps41-deltaN), Vps41 was tagged with a 3x-FLAG tag. In Vps18-deltaN the tag was placed instead to the C-terminus of Vps18 for more stability during purification. Therefore, it looks like that Vps41 is missing on the gel, since the subunit runs a little bit lower without tag and at the same size as Vps39. However, we observed densities for all six subunits as mentioned above.

[Editors’ note: further revisions were suggested prior to acceptance, as described below.]

Reviewer #1 (Recommendations for the authors):– While the authors are free to propose their favored interpretations, you need to separate speculation from evidence-based interpretation.

We have now moved any discussion and speculations to the Discussion section, including the presentation of Figure 4.

– Rigidity has a well-defined meaning as resistance to deformation induced by stress. Nothing in this study measures rigidity. The central core of HOPS may be conformationally stable to thermal fluctuations as an isolated complex, but there is no way to predict how resistant it would be to deformation when stressed by a process such as SNARE zippering.

We agree and removed or replaced the term “rigidity” throughout the text by “stable” as suggested by the reviewer. Among the 2.5 Mio individual particles, the core is the portion with the highest resolution, suggesting stability, or as we initially named it, rigidity. However, as the reviewer points out future studies have to test whether this part is indeed rigid.

– The authors propose that Vps33 lets go of the SNAREs during zippering but that due to other interactions, HOPS remains tightly enough associated with the SNAREs to promote fusion. That idea might be correct but no experimental support is provided, so it is a weak link in the model.

We present a working model based on previous data, which is – as every model – an interpretation of the data. Multiple studies identified binding sites of HOPS for N-terminal domains for vacuolar SNAREs (e.g. Laage et al., MBoC 2001; Lürick et al., JBC 2015; Song et al., *eLife* 2020). In addition, HOPS binds the entire SNARE complex (Lürick et al., JBC 2015; Krämer et al., MBoC 2011), maybe also via the N-terminal parts.

For HOPS, we propose that SNAREs remain bound via their N-terminal domains, even if Vps33 lets go. However, even the last step is only suggested from intermediates of structures by the Rizo and Hughson lab (Stepien et al., Science Advances 2022; Baker et al., Science 2015). We thus feel that our current model is not an overinterpretation, but a working model that is based on the structure we obtained – which has to be put to the test. We are as curious as you and the reviewers which part is further confirmed by data and which part requires revision.

– Another weak link is that the flexibility of the Ypt7-binding ends of HOPS are still challenging to reconcile with the idea that HOPS is a rigid scaffold.

This is another part that we can only speculate on, and we do not claim that we know the entire answer. We did not determine structures of Ypt7 with Vps41 or Vps39 as they did not withstand cryo-conditions and therefore used AlphaFold2 to model the interactions. Time will tell, how reliable this is, but the binding sites fit to all previous work by us and others (e.g. Plemel et al., Traffic 2011; Ostrowicz et al., Traffic 2010; Lürick et al., MBoC 2017).

The Kd between small GTPases and effectors is in the µM range, but may be further supported by other membrane binding sites. We tested, however, binding of HOPS to lipidated Ypt7-GTP on supported lipid monolayers. Here, HOPS required Ypt7 to bind to these membranes and we showed that it was sitting here upright with the Vps39 site (which has the higher affinity) proximal to the membrane (Füllbrunn et al., *eLife* 2021). As we obtained very consistent data, we concluded that the association of HOPS with Ypt7 is anything but floppy, even though we observe some dynamics. However, one has to keep in mind that, in this experiment, we only follow the mobility of the GFP associated with HOPS, and not HOPS itself between membranes.

We speculate that the Vps41-Ypt7 interface is also stable as HOPS drives tethering (Füllbrunn et al., *eLife* 2021; this study). And it may work this way, if Ypt7 binds to the membrane distal site of Vps41. Here, the hypervariable domain may be critical to allow Ypt7 to reach the site – very similar to the way TRAPPII can activate only Ypt31 (Bagde et al., Science Advances 2022).

We discuss part of this in the text and would like to keep it to this. We made sure that we here separate observations and models from interpretation.

– The bottom line is that as before, this study represents a major advance by describing a high-quality structure of HOPS. But the authors need to take seriously our request to separate speculation from interpretation, and to tone down the conclusion that this structure has revealed a new mechanism.

Please have a look. We think that we did.

Reviewer #2 (Recommendations for the authors):The authors clearly thought that his manuscript only needed a few minor tweaks, and didn't do a thorough re-write or come to grips with our central points.

We respectfully disagree. However, we have now further adapted our manuscript and think we made an even stronger effort in responding to all requests.

Ungermann and his colleagues are still confusing protein stability with rigidity. Well-resolved parts of structures must be stable, but may or may not be rigid, the former term referring to the energy needed to unfold, either locally or globally, the latter to the energy needed to deform conformation. The data give no indication on rigidity, while the authors strongly leans on rigidity in speculations about HOPS function. Ungermann hasn't addressed this at all, and the two references he cites in his reviewers responses are neither cited nor discussed in the revised manuscript.

Please see our response to reviewer #1.

The Abstract remains sadly misleading, stating that "HOPS is surprisingly rigid" and "The SNARE binding module is rigidly attached to the core.." (see above). The last sentence, "Our results explain HOPS dual functionality and unravel why tethering complexes are …essential…fusion" is therefore very inaccurate.

We adjusted the abstract accordingly.

There are no data on rigidity, as opposed to which regions of HOPS fold stably, but the manuscript persists in this vein. At the bottom of p6, they state "..it was unknown how tethering and fusion activities of HOPS are linked mechanistically. To clarify this…" but what follows addresses what a tethering structure of HOPS binding its Rab might look like through predictions and modeling. This doesn't address how "tethering and fusion activities of HOPS are linked mechanistically".

We agree that we did not test this, but rigidity was a term used to describe the part with the highest resolution among the 2.5 Mio particles analyzed by cryo-EM. We now went through the text and removed or replaced this term throughout.

They show that removing the β-propeller of Vps41 alters HOPS structure, but confuse this with an altered rigidity.

We agree. We observe an altered structure, which allows for tethering. The complex still has Vps33 bound. We interpret this as a less stable structure, and now rewrote the text accordingly.

The last paragraph before the Discussion is speculation and should be explicitly labeled as such and put in the Discussion. They present no evidence of HOPS continued association with the SNAREs as zippering proceeds. Their speculative model wasn't altered at all, and still shows Vps33 associating with the SNAREs as zippering progresses, though he acknowledges that this is very unlikely.

The reviewer is right, we moved this part to the discussion. As we wrote above, this is a working model and we do not go into the details here, how Vps33 may act. We feel that any further alterations of the model would not be more correct as it remains a working model, which is based on structure of intermediates (Munc18 with SNAREs; Vps33 with R and Q-SNAREs) and interactions between the N-terminal domains and HOPS – and our interpretation in light of our structure.

In short, the authors didn't take our reviews seriously. Re-reading the decision letter now, it was very clear that we wanted a thorough re-write, but we didn't get it.I suggest that we return the paper to the authors, and that they take our reviews to heart and thoroughly re-write this paper.

We hope that this reviewer acknowledges that we made an effort to include his/her suggestions throughout the text.

Reviewer #3 (Recommendations for the authors):Overall, I am still excited about this structure, and what it suggests for how tethering complexes can function. However, although some good changes have been made to the revised version, some of the conclusions are more speculative than the text would suggest.Some of the changes in the revision still do not adequately address all the reviewers' concerns. Most importantly, the idea of the central core being "rigid" is clearly supported by lack of other conformations in the cryoEM data, but that conclusion is still speculative with regard to the conformation during the fusion reaction in vivo, and needs to be softened a bit in the text. The fact that other conformations are not observed in the structure indicates that the purified complex under these conditions does not seem flexible. However, the structure may just represent the lowest energy, stable state. The data does not indicate that this region is "rigid" in a way that suggests that it cannot bend under other conditions, e.g. in vivo. Certainly future experiments will help understand this, but the readers should not come away with this idea as definitive.

We thank the reviewer for pointing this issue out to us. Our data revealed that the core of the HOPS complex, the backbone of Vps11 and Vps18 and the associated Vps16 and 33 subunits, had the highest resolution. In addition, the Vps11 and Vps18 subunits form a long hydrophobic interface. We agree that this does not indicate rigidity against distortion and thus removed this wording and the implications from the Results section. In the discussion, we speculate on the implications of the structure for HOPS function. We therefore moved also Figure 4 to the discussion as requested by the other reviewers as well and went through the entire text again to soften our interpretation and separate conclusions from speculations.

Although a little more information regarding other tethering complexes/membrane fusion machinery was added, it still is likely to be more relevant for a specialized readership, not a broad eLife audience.

We now extended this part further to describe the crosstalk between tethering complexes and membrane fusion. One main problem is, however, that functional data on tethering complexes is largely available from work on HOPS, whereas full reconstitution is mostly lacking for others. We acknowledge this and also mention thus the limitations in the discussion.

This previous concern was not addressed: "With regard to the structural analyses, it would be useful to speculate on the possible cause(s) for the difference between this structure and previous lower resolution structures, perhaps illuminating some additional (perhaps biologically relevant?) differences?" It is unclear how improvements and acceleration in the sample preparation helped yield a "rigid" molecule vs. previous studies.

We included now further explanations in the text regarding the sample preparation. There are several important improvements that helped us to solve the structure. One key factor was a protocol, where cells were cryo-lysed, affinity purification occurred via the FLAG tag on HOPS subunits, and the complex was eluted fast via the FLAG peptide from beads, whereas previously a TAP-tagged HOPS was used (where elution occurred over night). This is mentioned at the beginning of the results part and in detail described in the methods part. In brief, we accelerated all intermediate steps, improved concentration conditions and gel filtration and in the end had just a lot more intact particles to analyze. All these issues are mentioned, though we did not do a methods discussion. Of course, these improvements do not result in a different structure of the complex but more intact particles that we were able to analyze. When we previously analyzed HOPS, we had to use GRAFIX (an overnight glycerol gradient) to stabilize the complex, and the preparation took almost two day (Bröcker et al., 2012). Moreover, previously we and others used negative stain EM for analysis which has numerous well-known disadvantages for structure interpretation, for example flattening and strong inaccuracies for the euler angle assignment which easily can lead to wrong models and interpretations. Using cryoEM we can examine the complex in solution at very low temperatures, which diminishes drawbacks in negative stain.

"In Figure 3-S1 and the text, how good is the evidence that these Ypt7 binding sites predicted by Alphafold are accurate, especially as the location does not make a lot of biological sense with the location of the ALPS motif and membrane binding of Ypt7?" For this concern, I respectfully disagree that all Alphafold predictions are that accurate, especially if the biological conclusion is unclear, and this speculation should be better explained or the conclusion softened.

The reviewer is right, we can only interpret the AlphaFold model and need mutagenesis of the respective sites for confirmation of the Ypt7 binding sites. We noted however that these predictions match previous analyses (e.g. Plemel et al., Traffic 2011; Ostrowicz et al., Traffic 2010; Lürick et al., MBoC 2017), such as the deletion of the b-propeller of Vps41, which abolished tethering (Figure 3). We extended this part in the text and acknowledge the limitations of our interpretation.